# ChatEarthNet: A Global-Scale Image-Text Dataset Empowering Vision-Language Geo-Foundation Models

Zhenghang Yuan, Zhitong Xiong, Lichao Mou, and Xiao Xiang Zhu

Data Science in Earth Observation, Technical University of Munich, Munich 80333, Germany

**Correspondence:** Xiao Xiang Zhu (xiaoxiang.zhu@tum.de)

**Abstract.** The rapid development of remote sensing technology has led to an exponential growth in satellite images, yet their inherent complexity often makes them difficult for non-expert users to understand. Natural language, as a carrier of human knowledge, can bridge common users and complicated satellite imagery. Additionally, when paired with visual data, natural language can be utilized to train large vision-language foundation models, significantly improving performance in various tasks. Despite these advancements, the remote sensing community still faces a challenge due to the lack of large-scale, high-quality vision-language datasets for satellite images. To address this challenge, we introduce a new image-text dataset, providing high-quality natural language descriptions for global-scale satellite data. Specifically, we utilize Sentinel-2 data for its global coverage as the foundational image source, employing semantic segmentation labels from the European Space Agency's WorldCover project to enrich the descriptions of land cover types. By conducting in-depth semantic analysis, we formulate detailed prompts to elicit rich descriptions from ChatGPT. We then include a manual verification process to enhance the dataset's quality further. This step involves manual inspection and correction to refine the dataset. Finally, we offer the community ChatEarthNet, a large-scale image-text dataset characterized by global coverage, high quality, wide-ranging diversity, and detailed descriptions. ChatEarthNet consists of 163,488 image-text pairs with captions generated by ChatGPT-3.5 and an additional 10,000 image-text pairs with captions generated by ChatGPT-4V(ision). This dataset has significant potential for both training and evaluating vision-language geo-foundation models for remote sensing. The code is publicly available at https://doi.org/10.5281/zenodo.11004358 (Yuan et al., 2024b), and the ChatEarthNet dataset is at https://doi.org/10.5281/zenodo.11003436 (Yuan et al., 2024c).

# 1 Introduction

Land cover refers to the surface components of land, such as water body, tree, bare land, or developed area, providing the landscape patterns and features on the Earth's surface. A comprehensive understanding of global land cover holds significant relevance for international projects, such as the United Nations Framework Convention on Climate Change (UNFCCC) (Mora et al., 2014), as well as various applications, including urban planning, environmental assessment, disaster response, and economic development (García-Mora et al., 2012). Satellite imagery in the field of remote sensing is regarded as the ideal data for land cover monitoring, as it can provide an overview and repetitive observations of land cover (Franklin and Wulder, 2002). The Sentinel-2 mission (Drusch et al., 2012) has achieved great success in providing comprehensive satellite images that enable the Earth's surface to be monitored on a global scale. A thorough analysis of land cover using Sentinel-2 data not only enhances the understanding of ecosystems but also supports numerous practical applications, including natural resource management, agriculture, and food security (ESA, 2024a).

The rapid advancements in remote sensing technology have led to an exponential increase of tasks and benchmark datasets in semantic understanding of land cover types (Xiong et al., 2022). However, these tasks and datasets usually focus on an image-level and pixel-level understanding of land cover types and fail to convey rich semantic relationships and contextual information. Land cover maps, while detailed, can be challenging for non-expert users in terms of interpretation and effective utilization in practical applications. In contrast, natural language, with its rich semantic information, is regarded as a bridge between common users and complicated satellite imagery, serving as a crucial modality for understanding sophisticated machine learning systems (Lobry et al., 2021). For example, natural language is integrated into vision-language models on different tasks in a user-friendly manner, such as image captioning (Lu et al., 2017), visual question answering (Lobry et al., 2020; Yuan et al., 2022), visual grounding (Li et al., 2023a; Zhan et al., 2023), and referring image segmentation (Yuan et al., 2024a) in the remote sensing domain.

Despite the progress, the generalizability and performance of these vision-language models are limited by the small-scale models and training datasets. Recently, it has been shown that foundation models, pre-trained on extensive datasets, can be further fine-tuned for specific tasks across different domains, serving as versatile tools in artificial intelligence (Zhou et al., 2023; Xiong et al., 2024). Among foundation models, large language models and large vision-language foundation models have achieved significant advancements. For large language models, examples like ChatGPT (OpenAI, 2024) and LLaMA (Touvron et al., 2023) demonstrate notable progress. For large vision-language foundation models, CLIP (Radford et al., 2021), LLaVA (Liu et al., 2023b), MiniGPT-4 (Zhu et al., 2023), MiniGPT-v2 (Chen et al., 2023), and Qwen-VL (Bai et al., 2023) have revolutionized the computer vision community. These models are equipped with billions of parameters and trained on vast amounts of image-text data, offering a substantial improvement over traditional, small-scale models in the zero-shot transfer ability across various tasks (Zhang et al., 2024; Liu et al., 2023a). The success of large vision-language foundation models indicates the crucial role of large-scale, semantically aligned image-text datasets in enhancing their versatile capabilities. For natural images, large vision-language foundation models can utilize web-scale image-text pairs available on the internet, where images are associated with corresponding relevant text. However, few pairs on the web provide detailed descriptions for satel-

**Dataset name**: UCM-Captions        **Number**: 2100 images and each image has 5 captions        **Caption annotation method**: manual annotation

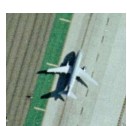
1. There is an airplane on the runway.
2. An airplane is taxiing on the runway.
3. It is an airplane taxiing on the runway.
4. There is an airplane taxiing on the runway.
5. An airplane is taxiing on the runway.

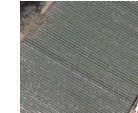
1. There is a piece of farmland.
2. There is a piece of cropland.
3. It is a piece of farmland.
4. It is a piece of cropland.
5. Here is a piece of farmland.

**Dataset name**: Sydney-Captions      **Number**: 613 images and each image has 5 captions       **Caption annotation method**: manual annotation

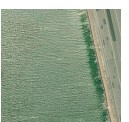
1. This is a part of deep green sparkling sea with a highway beside.
2. A part of ocean with deep green waters while a highway passed by.
3. This is a part of deep green sparkling sea while a highway passed by.
4. This is a part of deep green sparkling sea.
5. This is a part of deep green sparkling sea with a highway passed by.

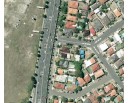
1. Some roads go through the residential area.
2. A residential area with houses arranged neatly and a highway beside this area.
3. A town with many houses arranged neatly while some roads go through this area.
4. A residential area and a wasteland are separated by a highway.
5. A residential area and a wasteland are separated by a highway.

**Dataset name**: RSICD          **Number**: 10,921 images and each image has 5 captions       **Caption annotation method**: manual annotation

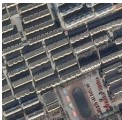
1. A playground and a parking lot are hemmed in an area with neatly-arranged buildings.
2. A regular court and a parking lot beside.
3. Many orderly buildings are around a playground.
4. A playground is semisurrounded by many orderly buildings.
5. A playground is next to a large piece of tall buildings.

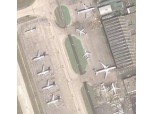
1. Some planes are parked in an airport.
2. The airport here is full of airplanes and containers.
3. The airport here is full of airplanes and containers.
4. Some planes are parked in an airport.
5. Some planes are parked in an airport.

**Dataset name**: NWPU-Captions     **Number**: 31,500 images and each image has 5 captions      **Caption annotation method**: manual annotation

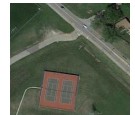
1. There are two tennis courts on the vacant lot.
2. Tennis court built on a lawn surrounded by road.
3. The tennis court is on the grass next to some trees and roads.
4. The tennis courts are surrounded by grass.
5. The tennis court is on a green meadow.

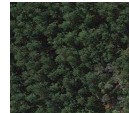
1. The forest has a lot of dense dark green trees.
2. This is a dense forest.
3. Many green trees are in a forest.
4. The forest is full of green trees.
5. There are many green trees in a forest.

**Dataset name**: RSICap        **Number**: 2,585 image-text pairs       **Caption annotation method**: manual annotation

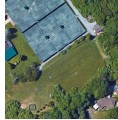
This is a high-resolution aerial image displaying a tennis court. Located in the upper left corner of the image are six tennis courts, two of which are partially visible. The tennis court surface is painted blue and has white markings. There is a large expanse of well-maintained grass and trees next to the tennis courts. In the lower right corner of the image, there is a building with a brown roof. Additionally, there is a building with a blue roof on the left side of the tennis courts.

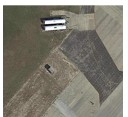
This is a high-resolution satellite image showing an airport, as a partially visible plane can be seen at the bottom of the image. On the left side of the image is a large grassy area with two white cars parked on it. On the right side of the image is a large open space, which is likely the airport runway.

**Dataset name**: RS5M         **Number**:  5 million images with captions        **Caption annotation method**: filtering public image-text datasets and captioning remote sensing datasets by BLIP2

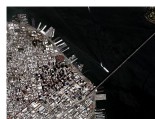
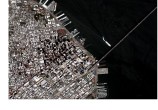
a satellite view of a large city in the water

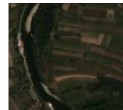
the small town with the dirt roads is shown in this satellite image

**Dataset name**:  SkyScript       **Number**:  2.6 million image-text pairs       **Caption annotation method**: linking remote sensing images with semantics in OSM via geo-coordinates

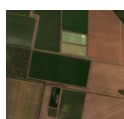
Single-object text: landuse of farmland, crop of cotton
Multi-object text:landuse of farmland with crop of cotton

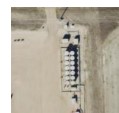
Single-object text: man made storage tank
Multi-object text: man made storage tank, surrounded by road of service; landuse of industrial with industrial oil

**Figure 1.** Comparative visualization of image-text pairs across UCM-Captions (Qu et al., 2016), Sydney-Captions (Qu et al., 2016), RSICD (Lu et al., 2017), NWPU-Captions (Cheng et al., 2022), RSICap (Hu et al., 2023), RS5M (Zhang et al., 2023), SkyScript (Wang et al., 2024) Datasets. For RS5M, only model-generated captions are shown.

lite images (Wang et al., 2024). This further confirms the need to construct large-scale, high-quality image-text datasets for remote sensing.

Although there have been several attempts to construct image-text datasets for remote sensing data, they still have limitations on the quality, quantity, and diversity of the provided image captions. Fig. 1 shows the comparative visualization along with the number and caption (description) annotation methods of existing available image-text pair datasets in the remote sensing domain, including the UCM-Captions (Qu et al., 2016), Sydney-Captions (Qu et al., 2016), RSICD (Lu et al., 2017),NWPU-Captions (Cheng et al., 2022), RSICap (Hu et al., 2023), RS5M (Zhang et al., 2023), and SkyScript (Wang et al., 2024) Datasets. These datasets range significantly in size, quality of caption, and annotation method. The dataset sizes vary from thousands to millions of image-text pairs. Although the RS5M and SkyScript datasets use algorithms to generate captions automatically and reach quantities in the millions, their text descriptions lack detail and only provide basic information. Similarly, smaller datasets like UCM-Captions, Sydney-Captions, RSICD, and NWPU-Captions predominantly feature simple captions, often limited to a single sentence for each caption. Though five captions are provided per image, the descriptions tend to be very similar or even identical. This simplicity and redundancy are disadvantages of these datasets. RSICap dataset stands out for its detailed manual annotation, but the quantity is limited, with only 2,585 image-text pairs. This is because it is manually annotated, a time- and labor-consuming process, making large-scale dataset generation difficult. In conclusion, these datasets suffer from limitations, with none of them encompassing both a large quantity of satellite images with global coverage and high-quality descriptions.

Our motivation is to construct a large-scale image-text dataset with global coverage that not only meets the semantic richness required for training large vision-language foundation models but also extends the understanding of satellite imagery to common users. For the data sources, we utilize Sentinel-2 data due to its practicality and accessibility. For the source of semantic information in Sentinel-2 data, we choose land cover maps from the European Space Agency's (ESA) WorldCover project (Zanaga et al., 2021). Leveraging Sentinel-2 data and the corresponding land cover maps, we aim to construct a global-scale, high-quality image-text dataset, which is essential for training large vision-language foundation models. However, it is challenging to manually annotate Sentinel-2 data on a large scale with high quality. This is mainly because manually annotating large datasets is time- and labor-consuming; the low resolution of Sentinel-2 images also makes it challenging to distinguish land cover types.

In this study, we introduce an automated processing framework for generating descriptions of satellite images, leveraging the powerful language generation capability of ChatGPT (OpenAI, 2024). Through the design of effective prompts, this framework can make use of ChatGPT to yield high-quality, detailed descriptions for Sentinel-2 imagery at a global scale. By integrating rich natural language descriptions with global satellite imagery, the proposed dataset fills in the interpretability gap between complex satellite imagery and common users. To further improve the quality of the dataset, we conduct a manual validation process to check the caption's correctness and quality. In summary, we offer the community ChatEarthNet, a large-scale image-text dataset with global coverage, high quality, wide-ranging diversity, and detailed descriptions. The vast amount of geotagged image-text pairs in ChatEarthNet is essential for training vision-language geo-foundational models, which are specifically designed to process and analyze geospatial satellite data.

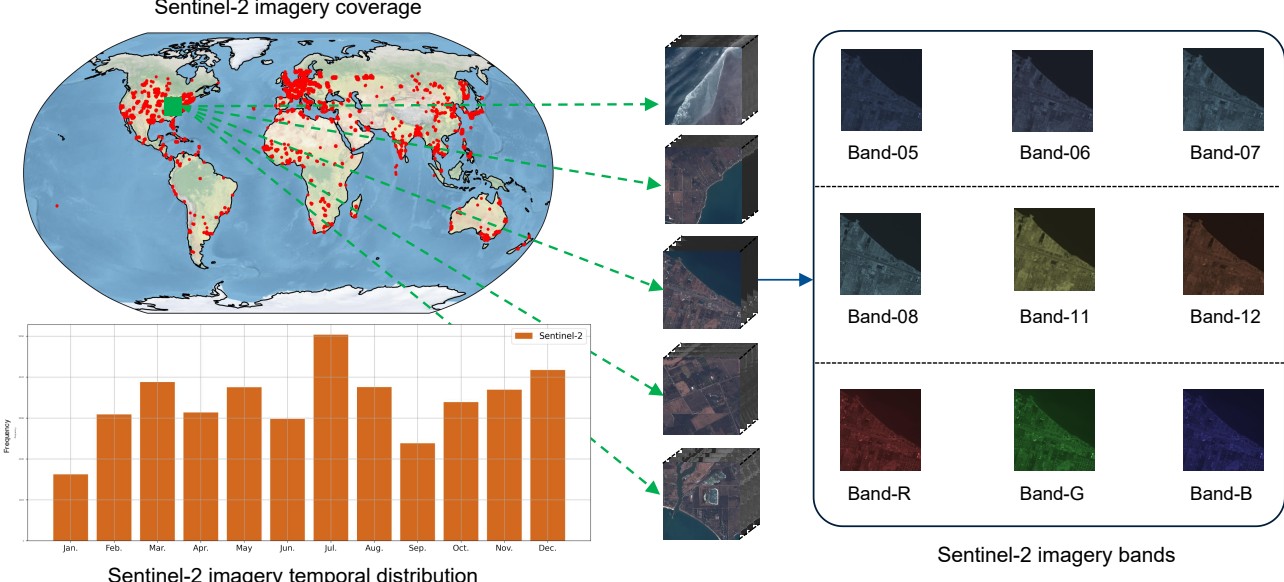

**Figure 2.** Statistics derived from Sentinel-2 data used in the ChatEarthNet dataset. The upper-left part of the figure displays the geographical distribution of the Sentinel-2 data used in the ChatEarthNet dataset. The lower-left part shows the temporal distribution of the Sentinel-2 data used. The right part visualizes some examples of the images and the nine spectral bands used in the dataset.

## 2 Dataset and methodology

The ChatEarthNet dataset is built upon the Sentinel-2 data (Drusch et al., 2012) with global coverage and the fine-grained land cover product from the ESA's WorldCover project (Zanaga et al., 2021). For each Sentinel-2 image, the overall dataset construction process contains the following steps: 1) we analyze its land cover distributions based on the WoldCover product; 2) we design sophisticated prompts based on the land cover distribution; 3) we generate descriptive texts based on prompts, using two versions of ChatGPT. This approach ensures that each description accurately reflects the visual data, providing a rich semantic description of the satellite imagery. Finally, the manual verification and correction of generated texts further improve the dataset's accuracy and quality. In this paper, ChatGPT-3.5 refers to the model gpt-3.5-turbo and ChatGPT-4V refers to the model gpt-4-vision-preview.

### 2.1 Sentinel-2 data in ChatEarthNet

Sentinel-2 (ESA, 2024b) provides global-scale optical imagery that captures a wide array of spectral bands, with a spatial resolution ranging from 10 to 60 meters. The spectral range of Sentinel-2 data is specifically tailored to monitor land cover types (Karra et al., 2021), making it invaluable for applications like agricultural monitoring and forestry management. Regarding Sentinel-2 images, we follow the sampling strategy used in the SatlasPretrain dataset (Bastani et al., 2023). Specifically, we

use the Sentinel-2 images collected in SatlasPretrain as the foundation to build the image-text dataset. This subsection details the characteristics of Sentinel-2 data used in the ChatEarthNet dataset.

1. **Global distribution**: The ChatEarthNet dataset is designed to capture a detailed description of the land cover, with its images spanning all continents except Antarctica and encompassing major urban centers, shown in the upper-left part of Fig. 2. The global distribution ensures diverse landscapes and urban areas, enriching the dataset with a variety of visual characteristics relevant to different geographical locations.

2. **Temporal coverage**: The temporal distribution of images is a critical aspect of the dataset. As illustrated in the bottom-left side of Fig. 2, the ChatEarthNet dataset includes Sentinel-2 images captured throughout different months to ensure that they cover different seasons on the Earth's surface. This temporal diversity allows the dataset to provide a more comprehensive appearance of different land cover types.

3. **Image size**: The spatial size of Sentinel-2 images in the ChatEarthNet dataset is $256 \times 256$ pixels. There is a total of 163,488 images in the dataset, providing extensive coverage across the world and enabling analysis and applications in various remote sensing tasks.

4. **Spectral band**: Sentinel-2 imagery is rich in spectral information, and the ChatEarthNet dataset includes nine specific bands from the S2A sensor, as shown in the right part of Fig. 2. These bands are band 5, band 6, band 7, band 8, band 11, band 12, along with the red, green, and blue (RGB) bands. The selected bands offer a detailed spectral resolution that captures a broad range of wavelengths, providing insights into different physical properties of the land cover.

Sentinel-2 image and land cover map pairs

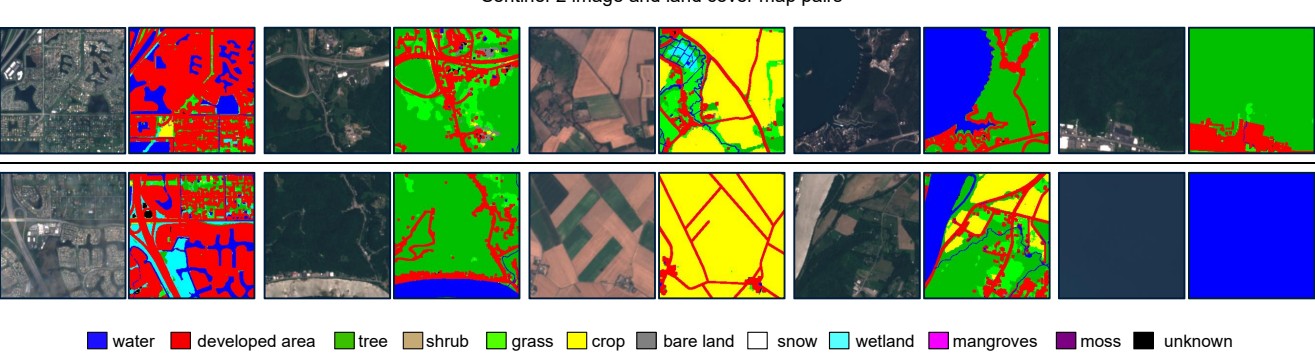

water developed area tree shrub grass crop bare land snow wetland mangroves moss unknown

**Figure 3.** Sentinel-2 image and land cover map pairs. The land cover maps, sourced from the WorldCover product, showcase various land cover types.

## 2.2 Land cover map from WorldCover product

As depicted in Fig. 3, the ChatEarthNet dataset leverages the WorldCover product 2020 version (Zanaga et al., 2021) to obtain semantic information, which provides global-scale land cover maps. These maps describe the land cover at 10 meters

resolution. Specifically, we utilize 11 different land cover classes (Bastani et al., 2023), including "water", "developed area", "tree", "shrub", "grass", "crop", "bare land", "snow", "wetland", "mangroves", and "moss". These land cover types offer a detailed categorization encompassing natural and urban-related landscapes, providing critical semantic information to generate detailed descriptions. The integration of WorldCover with Sentinel-2 data provides a robust foundation for our image-text dataset. By doing so, we can take advantage of both the global-scale satellite images and detailed land cover semantics.

## 2.3 Large language models for description generation

Among the numerous large language models that have been developed, ChatGPT is distinguished by its exceptional performance. Its proficiency in textual understanding and production makes it a valuable tool for textual analysis and description generation. Thus, in this work, we adopt two large language models, ChatGPT-3.5 and ChatGPT-4V(ision) to generate two different versions of image-text datasets.

While both ChatGPT-3.5 and ChatGPT-4V represent significant advancements in the field of large language models, they exhibit differences in their performance and capabilities. Compared with ChatGPT-3.5, ChatGPT-4V can process not only text but also visual inputs, thereby enhancing its contextual comprehension of the shapes and spatial distributions of land cover types in images. Moreover, ChatGPT-4V demonstrates improved performance in terms of accuracy, coherence, and the ability to handle sophisticated prompts. However, considering the Application Programming Interface (API) prices, ChatGPT-4V is much more expensive than ChatGPT-3.5. Additionally, by February 2024, for usage tier 1, ChatGPT-4V has a limit of 500 requests per day, while ChatGPT-3.5 has a limit of 10,000 requests per day. If processing one image (image represented by text for GPT-3.5) requires a single request, this means that GPT-3.5 can handle 10,000 images per day, while GPT-4V is limited to processing just 500 images daily. Considering cost and efficiency, we utilize ChatGPT-3.5 for generating descriptions for the complete dataset, comprising 163,488 image-text pairs, and randomly select a subset of 10,000 Sentinel-2 images for description generation using ChatGPT-4V, resulting in 10,000 image-text pairs.

## 2.4 Prompt design

In this section, we detail the prompt designs for caption generation using ChatGPT. Given that ChatGPT is predominantly trained on commonly available natural images, its direct application to satellite images may not yield optimal results. To address this, we carefully design prompts that embed semantic information from the land cover maps for the caption generation. This allows the large language models to utilize the provided context to generate precise and semantically rich descriptions for satellite images. Although the underlying concept is straightforward, it requires careful designs to compensate for the inherent limitations of the current versions of ChatGPT. These limitations include challenges with instructions following, where large language models may not strictly follow the given instructions in the prompt. The other limitation is the well-known hallucination problem, where large language models may output plausible but factually incorrect or nonsensical information. To alleviate these issues, we carefully design the prompts and instructions to guide ChatGPT toward generating reliable and contextually appropriate descriptions.

The term "prompt" in large language models like ChatGPT refers to the input provided to the model to generate a response. System prompt and user prompt serve different functions, as illustrated below:

1. **System prompt**: System prompt typically refers to the initial instructions set by developers for configuring large language models. Its purpose is to establish the ground rules or guidelines for the following conversation, including setting the tone, style, or scope of the responses to standardize the large language model's behavior.

2. **User prompt**: User prompt is the actual question or statement input by the user, seeking a response from large language models. This is the variable part of the interaction that can differ with each user. Context information can be part of the user prompt to provide background details necessary for large language models to generate relevant and coherent responses. It provides additional information to give large language models a better understanding of the current topic.

### 2.4.1 Prompt design for ChatGPT-3.5

We aim to generate captions, i.e., natural language descriptions, for Sentinel-2 satellite images. To provide sufficient semantic information, we leverage the geographic-aligned land cover maps derived from the WorldCover product. Given the corresponding land cover map, we generate textual descriptions based on the proportions of different land cover types. Since ChatGPT-3.5 can only accept text instructions as input, we need to extract the semantic information from land cover maps and provide it to ChatGPT-3.5 in the textual form. The designed prompt for ChatGPT-3.5 is presented in the following block.

---

**Prompt design for ChatGPT-3.5**

messages = ["Analyze the provided image as an AI visual assistant. The following contexts are provided. The overall land cover distributions from most to least are: <output of Algorithm 1>

You are an AI visual assistant who can help describe images based on the given contexts. Please write the description in a paragraph, and avoid saying other things. The following constraints should be obeyed:

1. Describe the image in the order of the spatial distributions presented in the given contexts. Link descriptions of different parts to make the overall image description more fluent.

2. Describe the dominant land cover type in the image and its spatial locations.

3. Describe the land cover types in each part of the image in descending order of their coverage areas.

4. Diversify descriptions related to portions in each paragraph.

5. Summarize the main theme of the image in the final sentence.

6. Describe it objectively; do not use words: 'possibly', 'likely', 'perhaps', 'context', 'segmentation', 'appear', 'change', 'transition', 'dynamic', or any words with similar connotations."]

---

**Algorithm 1** Generating the prompt for land cover proportion

**Data:** Land cover map $Y$

**Result:** Generated prompt containing land cover proportions for $Y$

1 **Function** `GeneratePrompt(`$Y$`)`:

2  Compute the overall proportions of different land cover types in $Y$

   Generate a prompt describing the overall land cover proportions

   Split $Y$ into five patches: $Y_{tl}, Y_{tr}, Y_{bl}, Y_{br}, Y_m$

   **foreach** *patch $Y_i$* **do**

3    Compute the number of land cover types $n_i$ in $Y_i$

     Compute the proportion of each land cover type in the patch

     Sort the land cover types according to proportions in a descending order

     Select three (if $n_i$ is less than three, select $n_i$) land cover types from most to least

     Generate a prompt for the patch describing the portions of all the selected land cover types

4  Concatenate all prompts

   **return** concatenated prompts

---

The prompt comprises two elements: the system prompt to guide the response style and set constraints to ChatGPT, and the user prompt containing context derived from land cover maps using **Algorithm 1**. The system prompt includes a set of explicit constraints to ensure the generated descriptions are fluent, accurate, and unbiased. Specifically, we force ChatGPT to generate fluent descriptions and focus more on the spatial locations and portions of different land cover types. We also encourage ChatGPT to describe objectively and avoid the use of subjective words.

For the user prompt, we extract the semantic information from land cover maps, where each pixel represents a land cover type. Specifically, in **Algorithm 1**, we first calculate the overall proportions of different land cover types and generate a prompt describing the overall land cover proportions. Subsequently, we split the land cover map $Y \in \mathbb{R}^{256 \times 256}$ into four non-overlapping patches of equal size, each being $128 \times 128$. The top-left patch, denoted as $Y_{tl} \in \mathbb{R}^{128 \times 128}$, extends from indices 0 to 127 in both row and column directions. The top-right ($Y_{tr}$), bottom-left ($Y_{bl}$), and bottom-right ($Y_{br}$) quadrants are similarly demarcated. Additionally, we extract a middle patch $Y_m$, also $128 \times 128$ in size, centered within the map, with indices ranging from 64 to 191 in both the row and column directions, aligning with the midpoint of the land cover map.

For each patch $Y_i$, we calculate the proportion of each land cover within that patch relative to its total pixel count. We then rank these land cover types by their proportional presence and select the top three to represent the primary land cover types of the patch. In cases where a patch contains fewer than three land cover types, we select all available types. This selection process is employed because ChatGPT-3.5 tends to generate verbose descriptions when presented with abundant prompts. Limiting the information to three main land cover types ensures more focused descriptions, and can avoid unnecessary lengthy captions. After determining the primary land cover types for all five patches in a land cover map $Y$, we concatenate their proportions and

the overall proportions to formulate the final prompt. This tailored prompt enables ChatGPT-3.5 to generate accurate, detailed, and coherent descriptions of Sentinel-2 satellite imagery.

### 2.4.2 Prompt design for ChatGPT-4V

---

**Prompt design for ChatGPT-4V**

messages = [ "You are an AI visual assistant that can analyze the given image. In the image, different colors represent different land cover types. The color for the land cover dictionary is: '[0, 0, 255] (blue): water; [255, 0, 0](red): developed area; [0, 192, 0] (dark green): tree; [200, 170, 120] (brown): shrub; [0, 255, 0] (green): grass; [255, 255, 0] (yellow): crop; [128, 128, 128] (grey): bare; [255, 255, 255] (white): snow; [0, 255, 255] (cyan): wetland; [255, 0, 255] (pink): mangroves; [128, 0, 128] (purple): moss.' You will be provided with four independent images at once.

For the first/second/third/fourth images, the distribution of each land cover type is: <output of Algorithm 2> For the first/second/third/fourth images, the spatial distribution of the image is: <output of Algorithm 3> You are given four independent images, describe in long sentences for each image separately using four paragraphs, and avoid saying other things. The following constraints should be obeyed:

1. Do not use color-related words; treat the color as the land cover type directly.

2. Generate the four descriptions separately; do not add connections between them.

3. When describing water, developed, and crop areas, incorporate shape descriptors.

4. Double-check all the presented land cover types based on the distribution of each land cover type. If some land covers are not presented, do not mention them.

5. Describe it objectively; do not use words: 'possibly', 'likely', 'perhaps', 'color dictionary', 'appear', 'change', 'transition', 'dynamic', or any words with similar connotations.

6. Double-check the shape and location of the developed area, water course, grass, tree, shrub, wetland, and crop areas based on the given image if they are present.

7. Consider the spatial statistics as a unified image without breaking them down into individual spatial distributions and land cover proportions when describing the overall scene.

8. Describe each land cover separately for each given image, and then describe the main theme of each given image."]

190

---

The prompt design for ChatGPT-4V is presented in the text block above. Like the prompt for ChatGPT-3.5, this prompt contains a system prompt and a user prompt. However, the system prompt for ChatGPT-4V differs from that used for ChatGPT-3.5, as ChatGPT-4V is capable of processing the land cover map as an image directly. Given that the land cover map is

essentially a segmentation map where each color represents a land cover type, this key information is provided to ChatGPT-4V through the system prompt. To enhance the accuracy and detail of descriptions, we also define several guides and constraints in the system prompt. Moreover, considering the API request limit of ChatGPT-4V, we put four images into one request to generate descriptions more efficiently. While ChatGPT-4V can handle image inputs, it still requires specific guidance to accurately interpret segmentation maps from a remote sensing perspective. Hence, the user prompt is supplemented with semantic information extracted from the land cover maps using **Algorithm 2** and **Algorithm 3**.

Similar to the process described in **Algorithm 1**, we split the land cover map $Y \in \mathbb{R}^{256 \times 256}$ into five different patches: top-left $Y_{tl}$, top-right $Y_{tr}$, bottom-left $Y_{bl}$, bottom-right $Y_{br}$ and middle $Y_m$ patches, each being $128 \times 128$. As shown in **Algorithm 2**, for each patch, we calculate the proportion of each land cover within that patch relative to its total pixel count. Different from **Algorithm 1**, we provide the proportion information of all land cover types (instead of three main land cover types) in each patch to ChatGPT-4V. The reason is that ChatGPT-4V is more powerful and can process all information to generate detailed descriptions without unnecessarily lengthy descriptions. In **Algorithm 3**, we aim to calculate the distribution of each land cover type across the five patches. For each land cover type $L_j$, we first calculate the number of pixels for $L_j$ in the land cover map $Y$, represented by $N_j$. Subsequently, for each patch $Y_i$, we calculate the pixel count of $L_j$ in $Y_i$, denoted as $n_{ji}$. The spatial distribution is evaluated using the ratio $\frac{n_{ji}}{N_j}$, which quantifies the presence of $L_j$ in each patch relative to its overall occurrence. After computing the spatial distribution of $L_j$ across all patches, we concatenate prompts for all land cover types. These prompts, derived from calculations in both algorithms, are put into the final text prompt. This text prompt and the land cover map as visual input are then provided to ChatGPT-4V to generate descriptions.

---

**Algorithm 2** Generating the prompt for land cover proportion in each patch

---

**Data:** Land cover map $Y$

**Result:** Generated prompt containing land cover proportions for five patches of $Y$

5 **Function** `GeneratePrompt(Y)`:

6     Split $Y$ into five patches: $Y_{tl}, Y_{tr}, Y_{bl}, Y_{br}, Y_m$

        **foreach** *patch $Y_i$* **do**

7             Compute the proportion of each land cover type in the patch

            Generate a prompt for the patch describing the portions of all land cover types

8     Concatenate prompts for all patches

        **return** concatenated prompts

---

---
**Algorithm 3** Generating the prompt for the spatial distribution of each land cover type
---
**Data:** Land cover map: $Y$

**Result:** Spatial distribution of each land cover type across patches.

**9 Function** `GeneratePrompt(`$Y$`)`:

**10**     Split $Y$ into five patches: $Y_{tl}, Y_{tr}, Y_{bl}, Y_{br}, Y_m$

        **foreach** *land cover type $L_j$* **do**

**11**             Calculate the number of pixels for land cover type $L_j$ in $Y$, denoted as $N_j$

            **foreach** *patch $Y_i$* **do**

**12**                 Calculate the number of pixels for land cover type $L_j$ in $Y_i$, denoted as $n_{ji}$

                Calculate the spatial distribution via $\frac{n_{ji}}{N_j}$

**13**             Generate a prompt describing the spatial distribution of $L_j$ across all patches.

            Concatenate prompts for all land cover types

**14**     **return** concatenated prompts

---

## 2.5 Manual verification

To further improve the quality of the dataset, we conduct a manual validation process to check the caption's correctness and quality. Considering the efficiency and cost savings, we combine four images and the corresponding textual prompts in one request and provide them to ChatGPT-4V for caption generation. To avoid unexpected descriptions on comparisons between different images, we design prompts like "Generate the four descriptions separately; do not add connections between them" to guide the description generation process. Despite providing specific instructions for ChatGPT-4V to treat each image individually, it occasionally makes mistakes by describing comparisons between images. For instance, phrases such as "similar to other images" and "compared with previous images," need to be revised to eliminate comparisons. We therefore manually check all captions and refine comparison-related captions. For ChatGPT-3.5, we provide a single image (represented by text) in one request, which avoids the comparison issues. We manually inspected 10,000 image-text pairs from ChatGPT-3.5-generated captions, to ensure that there are no significant quality issues.

## 3 Dataset analysis and discussion

In this section, we present a comprehensive analysis of the ChatEarthNet dataset from different aspects. As we construct the dataset using ChatGPT-3.5 and ChatGPT-4V, we analyze and compare these two different versions to provide a clear overview and understanding of the ChatEarthNet dataset.

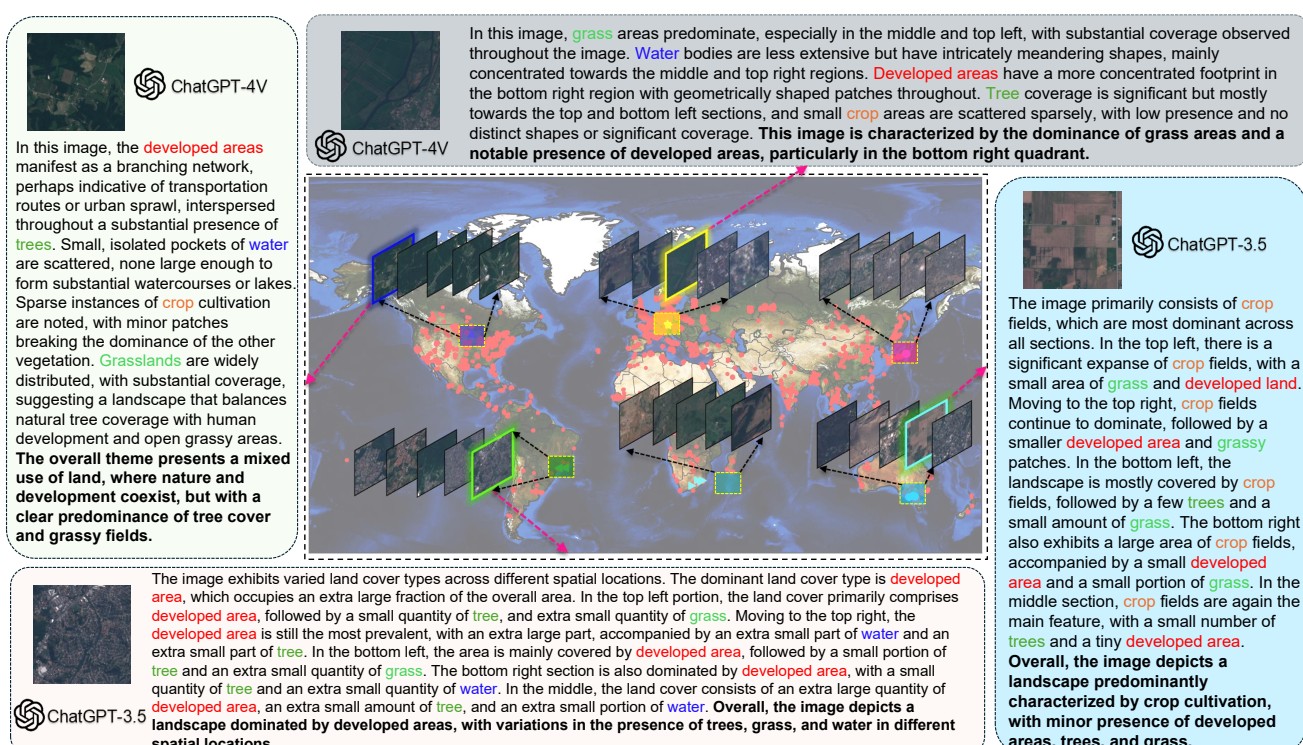

In this image, the developed areas manifest as a branching network, perhaps indicative of transportation routes or urban sprawl, interspersed throughout a substantial presence of trees. Small, isolated pockets of water are scattered, none large enough to form substantial watercourses or lakes. Sparse instances of crop cultivation are noted, with minor patches breaking the dominance of the other vegetation. Grasslands are widely distributed, with substantial coverage, suggesting a landscape that balances natural tree coverage with human development and open grassy areas. **The overall theme presents a mixed use of land, where nature and development coexist, but with a clear predominance of tree cover and grassy fields.**

In this image, grass areas predominate, especially in the middle and top left, with substantial coverage observed throughout the image. Water bodies are less extensive but have intricately meandering shapes, mainly concentrated towards the middle and top right regions. Developed areas have a more concentrated footprint in the bottom right region with geometrically shaped patches throughout. Tree coverage is significant but mostly towards the top and bottom left sections, and small crop areas are scattered sparsely, with low presence and no distinct shapes or significant coverage. **This image is characterized by the dominance of grass areas and a notable presence of developed areas, particularly in the bottom right quadrant.**

The image primarily consists of crop fields, which are most dominant across all sections. In the top left, there is a significant expanse of crop fields, with a small area of grass and developed land. Moving to the top right, crop fields continue to dominate, followed by a smaller developed area and grassy patches. In the bottom left, the landscape is mostly covered by crop fields, followed by a few trees and a small amount of grass. The bottom right also exhibits a large area of crop fields, accompanied by a small developed area and a small portion of grass. In the middle section, crop fields are again the main feature, with a small number of trees and a tiny developed area. **Overall, the image depicts a landscape predominantly characterized by crop cultivation, with minor presence of developed areas, trees, and grass.**

The image exhibits varied land cover types across different spatial locations. The dominant land cover type is developed area, which occupies an extra large fraction of the overall area. In the top left portion, the land cover primarily comprises developed area, followed by a small quantity of tree, and extra small quantity of grass. Moving to the top right, the developed area is still the most prevalent, with an extra large part, accompanied by an extra small part of water and an extra small part of tree. In the bottom left, the area is mainly covered by developed area, followed by a small portion of tree and an extra small quantity of grass. The bottom right section is also dominated by developed area, with a small quantity of tree and an extra small quantity of water. In the middle, the land cover consists of an extra large quantity of developed area, an extra small amount of tree, and an extra small portion of water. **Overall, the image depicts a landscape dominated by developed areas, with variations in the presence of trees, grass, and water in different spatial locations.**

**Figure 4.** An overview of the ChatEarthNet dataset. We randomly select image-text samples from four different locations. The left and top sides display the descriptions generated by ChatGPT-4V, while the right and bottom sides show two samples produced by ChatGPT-3.5. We use different colors to highlight the words of different land cover types.

## 3.1 Dataset overview

In Fig. 4, we present four different image-text pairs from four regions of the Earth, illustrating that images from different geographical locations exhibit unique characteristics. The diversity in land cover distributions across these images is evident. The accompanying texts accurately reflect the quantity and spatial distribution of the various land cover types observed.

In Table 1, we present the number of Sentinel-2 images used for generating captions, along with the corresponding numbers of captions generated by ChatGPT-3.5 and ChatGPT-4V in the ChatEarthNet dataset. Specifically, we use 163,488 Sentinel-2 images and generate a long caption to accompany each image using ChatGPT-3.5. For the ChatGPT-4V version, we randomly select 10,000 Sentinel-2 images across the world and generate one detailed caption for each image. In terms of the number of image-text pairs, the ChatEarthNet dataset is not the largest dataset available, but it offers high-quality detailed land cover descriptions on a global scale. This makes it a solid foundation for training vision-language geo-foundation models in the field of remote sensing.

**Table 1.** The number of Sentinel-2 images used for generating captions, along with the corresponding numbers of captions generated by ChatGPT-3.5 and ChatGPT-4V.

| Subsets | Number of ChatGPT-3.5 Captions | Number of ChatGPT4-V Captions |
|---------|-------------------------------|-------------------------------|
| Train | 98,092 | 6000 |
| Val | 16,348 | 1000 |
| Test | 49,048 | 3000 |
| Sum | 163,488 | 10,000 |

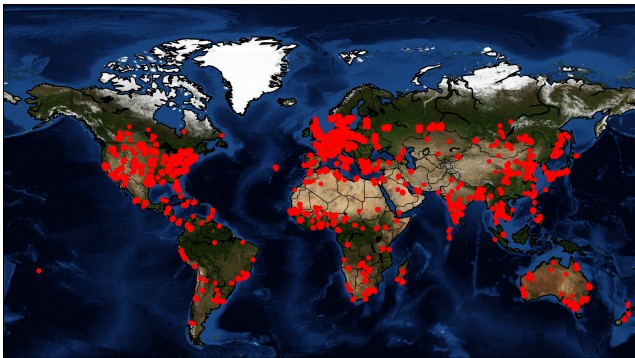

**Figure 5.** Geographical distribution of image-text pairs using ChatGPT-3.5.

**Figure 6.** Geographical distribution of image-text pairs using ChatGPT-4V.

### 3.2 Geographic coverage

Figs. 5 and 6 illustrate the geographical distribution of image-text pairs using ChatGPT-3.5 and ChatGPT-4V in the ChatEarth-Net dataset, respectively. From the two figures, we can see that the image-text pairs for both the ChatGPT-3.5 and ChatGPT-4V versions cover all continents except Antarctica. Compared to the image-text pairs using ChatGPT-4V, the geographical distribution of those using ChatGPT-3.5 is more dense, covering a wider range of areas. Nevertheless, 10,000 high-quality image-text pairs using ChatGPT-4V are sufficient for fine-tuning large vision-language models.

### 3.3 Word frequency

Figs. 7 and 8 illustrate the word clouds for captions generated by ChatGPT-3.5 and ChatGPT-4V, respectively. In the two figures, larger words indicate a higher frequency of occurrence. In Fig. 7, prominent words like "developed", "small", "medium", "grass", and "portion" indicate a focus on describing the content and scale of land cover types. Other significant words like "right" and "bottom" relate to specific locations in the image. In Fig. 8, the word cloud centers around "image" and "areas", indicating these are key themes in the generated captions. Adjacent to these are other significant words like "developed", "bottom", "water", "right" and "landscape", suggesting an emphasis on geographical features and the layout in the image.

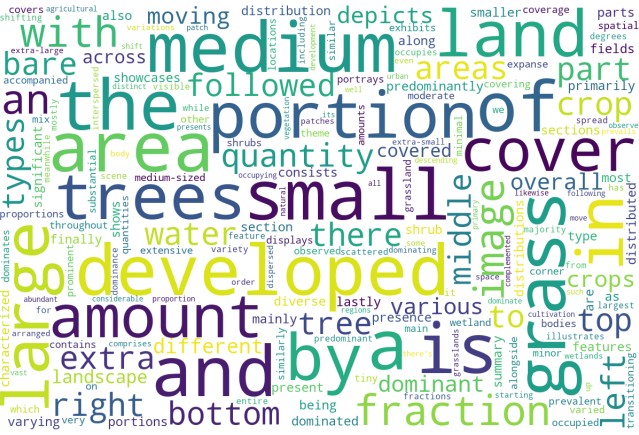

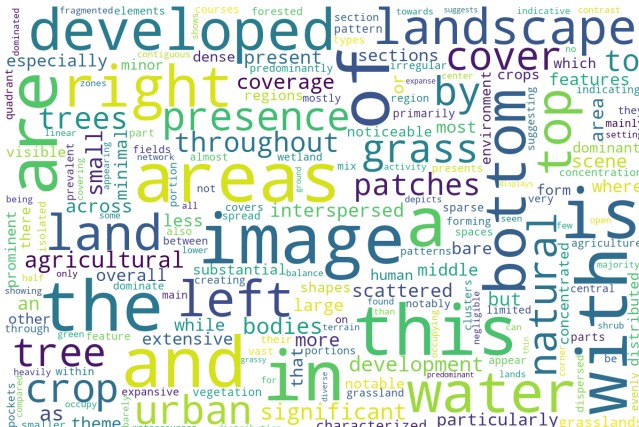

**Figure 7.** Word cloud for captions generated by ChatGPT-3.5.      **Figure 8.** Word cloud for captions generated by ChatGPT-4V.

Overall, the captions generated by ChatGPT-3.5 provide more straightforward descriptions focusing on the distribution and size of land cover types. The captions generated by ChatGPT-4V use more varied and descriptive language and showcase a more diverse vocabulary to describe the scale of land cover types and their layout.

Figs. 9 and 10 display histograms of the top 200 word frequencies for captions generated by ChatGPT-3.5 and ChatGPT-4V,
respectively. The x-axis represents individual words, and the y-axis represents the frequency. Both distributions are long-tailed, indicating that a minority of words are used frequently, while the majority appear infrequently. Comparing the two histograms, we observe that the descent from the most to least frequent words appears sharper in Fig. 9, while Fig. 10 exhibits a more gradual decline. This observation indicates that ChatGPT-4V employs a broader vocabulary to generate more diverse and higher-quality captions.

To better understand the differences in captions related to land cover types generated by ChatGPT-3.5 and ChatGPT-4V, we construct histograms to illustrate the frequencies of relevant words, as depicted in Figs. 11 and 12. The x-axis represents land cover types, and the y-axis represents the frequency. The histogram in Fig. 11 exhibits a clear long-tailed distribution, with "developed area", "grass", and "crop" being the most frequently mentioned land cover types. In Fig. 12, "developed area", "water", and "tree" are predominant land cover types. These differences reflect the different descriptive approaches and varied
geographical distributions in the two versions.

Figs. 13 and 14 illustrate the word frequencies related to quantity and shape for two versions of captions. The x-axis represents words related to quantity and shape, and the y-axis represents the frequency. The histogram for ChatGPT-3.5 shows a preference for terms like "small", "medium", "large", and "dominant" to describe land cover proportions. Meanwhile, ChatGPT-4V, as reflected in the histogram, employs a more diverse vocabulary, extending beyond common descriptors such as "small",
"large", and "dominant" to include high frequencies of "significant", "scattered", "minimal", "extensive", and "substantial". These words enrich the descriptions of land cover type shapes and patterns, indicating that captions of the ChatGPT-4V version leverage a broader vocabulary to describe the characteristics of the image.

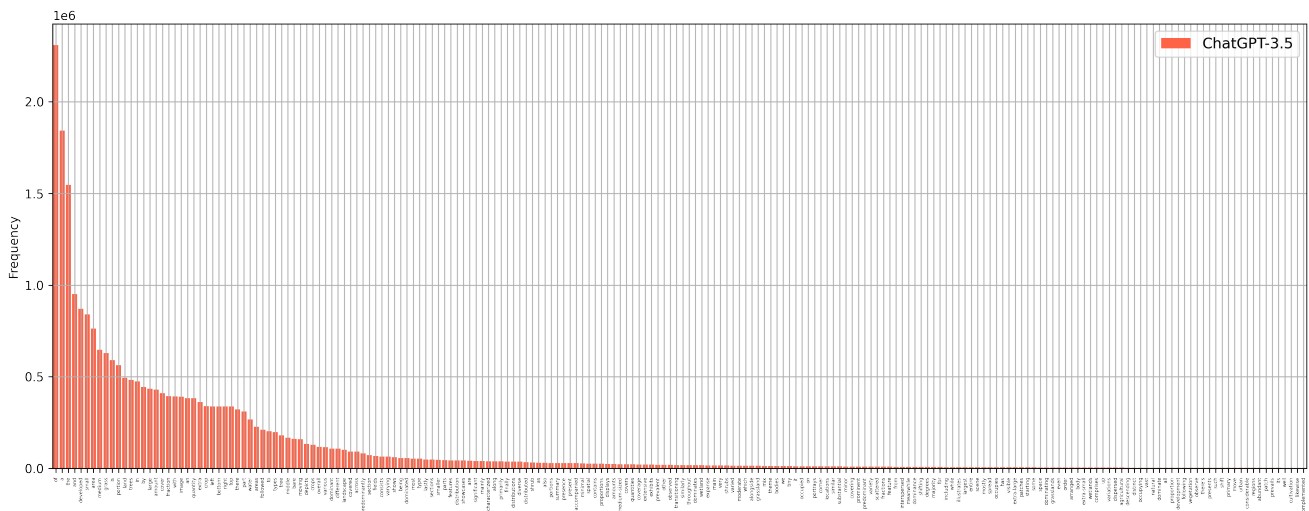

**Figure 9.** Histogram of word frequencies for captions generated by ChatGPT-3.5.

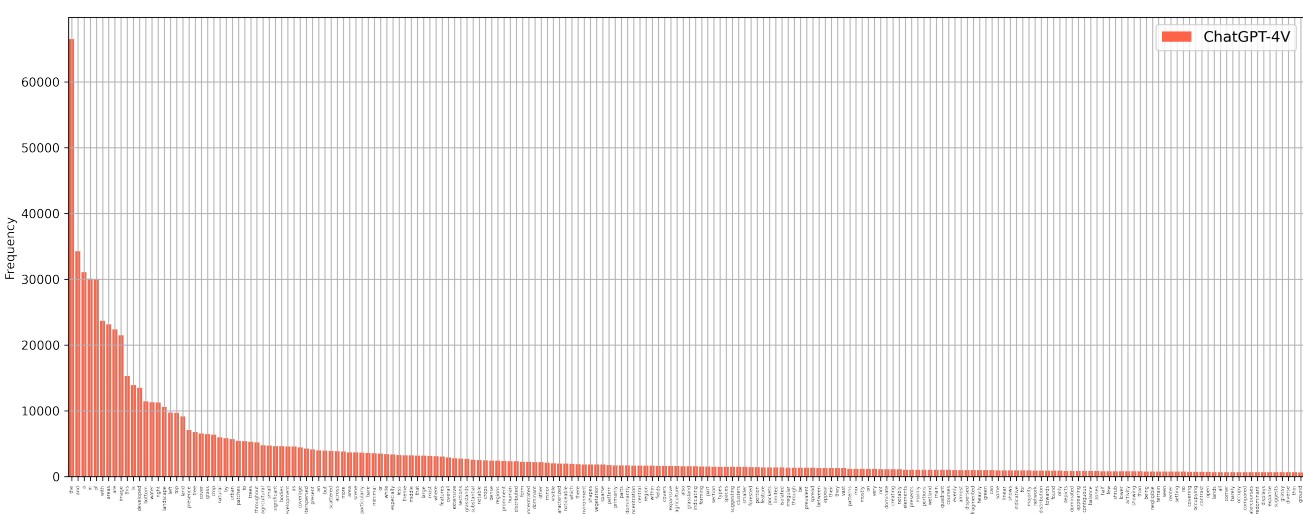

**Figure 10.** Histogram of word frequencies for captions generated by ChatGPT-4V.

## 3.4 Caption length

Fig. 15 presents a comparison of caption lengths generated by ChatGPT-3.5 and ChatGPT-4V, illustrated as the histogram.
The x-axis denotes caption length, and the y-axis represents the normalized frequency of captions at each length. Unlike most existing image-text datasets that typically provide brief annotations, the ChatEarthNet dataset stands out by offering comprehensive captions that provide detailed semantic insights into land cover types. The histogram for ChatGPT-4V, shown

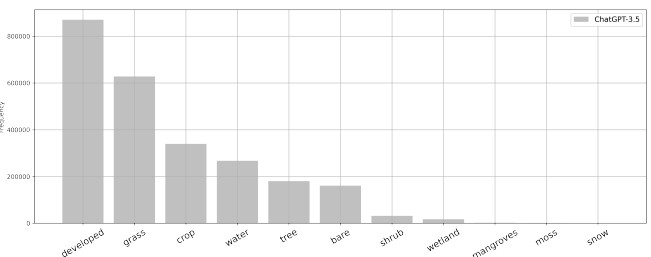

**Figure 11.** Histogram of word frequencies related to land cover types for captions generated by ChatGPT-3.5.

**Figure 12.** Histogram of word frequencies related to land cover types for captions generated by ChatGPT-4V.

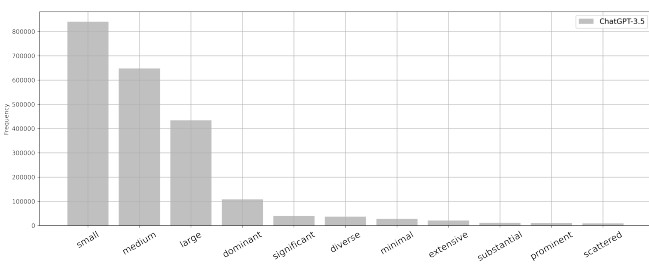
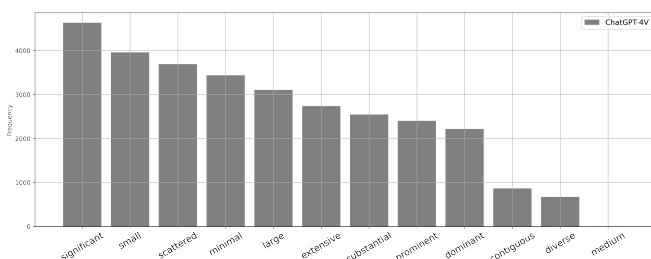

**Figure 13.** Histogram of word frequencies related to quantity and shape for captions generated by ChatGPT-3.5.

**Figure 14.** Histogram of word frequencies related to quantity and shape for captions generated by ChatGPT-4V.

in green, forms a Gaussian distribution with a mean value of around 90 words per caption. The histogram for ChatGPT-3.5, depicted in orange, also shows a Gaussian distribution but with a mean centered around 160 words, suggesting that captions generated by this version are generally longer. The reason is that ChatGPT-3.5 tends to elaborate on provided prompts by extending contextual cues, resulting in detailed descriptions that try to encompass various aspects of prompts. Conversely, ChatGPT-4V comprehensively grasps contextual information in prompts, enabling it to generate concise yet comprehensive descriptions. Additionally, ChatGPT-4V harnesses visual data (land cover maps), to achieve a more precise comprehension of spatial distributions of land cover types. As mentioned in the descriptions of Figs. 9 and 10, captions in the ChatGPT-4 version utilize a more diverse vocabulary. Consequently, the ChatGPT-4V captions manage to be more concise yet more varied.

## 3.5 Visualization and comparison

In Fig. 16, we showcase captions generated by ChatGPT-3.5 and ChatGPT-4V for a detailed comparison between the two versions. The caption from ChatGPT-3.5 provides a structured breakdown of the land cover types in five sections (top left, top right, bottom left, bottom right, and middle) of the image. This is a result of ChatGPT-3.5's inability to process image inputs directly, heavily relying on the given prompts. By doing so, these captions are structured, quantitative, and exhaustive, providing a balanced view of land cover types. In contrast, the caption from ChatGPT-4V adopts a holistic perspective, depicting land cover types in the context of the complete image rather than discrete sections. The language is descriptive and vivid,

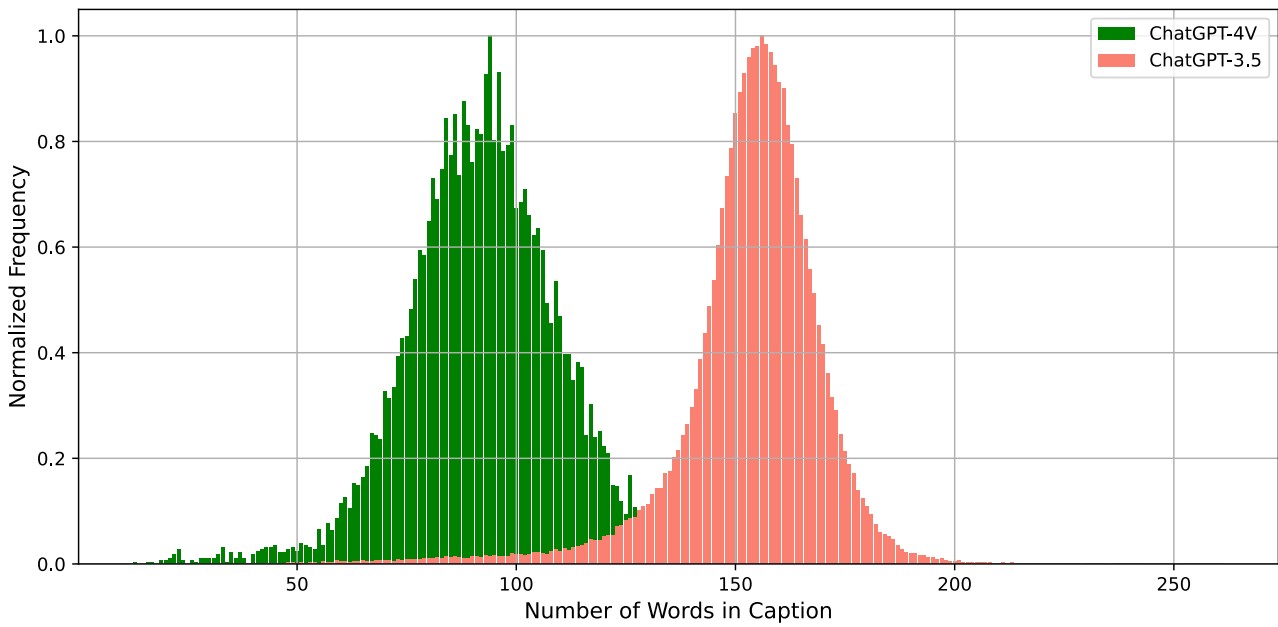

**Figure 15.** Histogram comparing caption lengths generated by ChatGPT-3.5 and ChatGPT-4V.

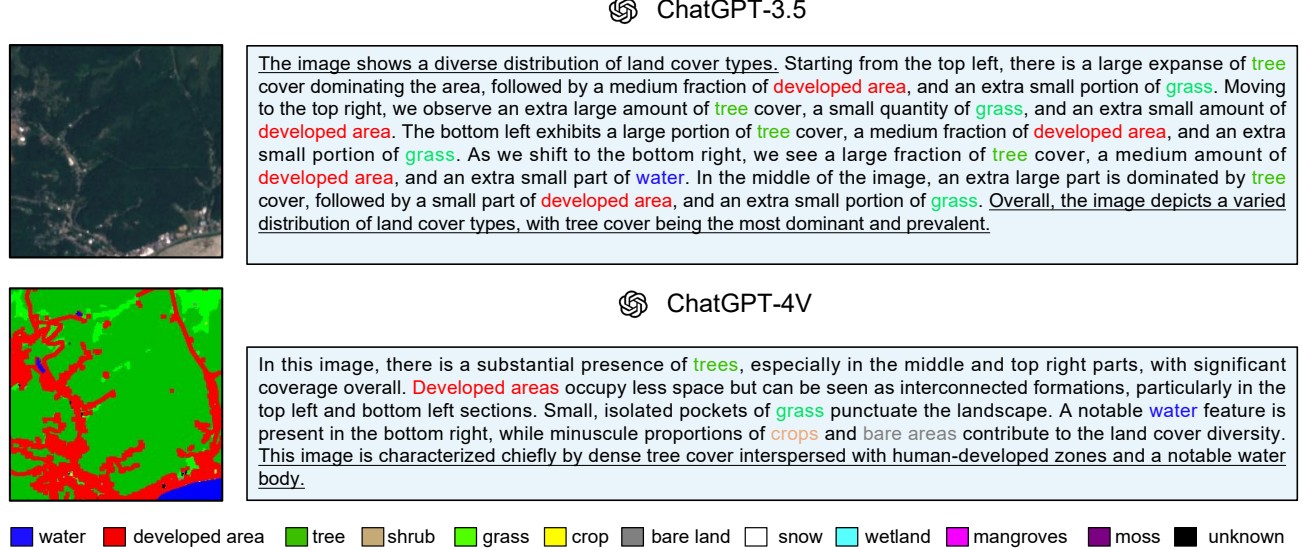

**Figure 16.** Sentinel-2 satellite image, its associated land cover map, and its corresponding captions generated by ChatGPT-3.5 and ChatGPT-4V.

emphasizing visually striking features and the general impression of the landscape. As ChatGPT-4V employs land cover maps as visual inputs, the generated captions offer a more comprehensive perspective, emphasizing the overall visual impact. While both captions offer different interpretations, each remains factually correct. The captions in the ChatEarthNet dataset can be valuable resources for the advancement of vision-language geo-foundation models in the field of remote sensing.

### 3.6 Evaluation of existing multimodal large language models using ChatEarthNet

To demonstrate the effectiveness of ChatEarthNet in evaluating multimodal large language models, we conduct benchmarking experiments using a range of existing models. Given that ChatEarthNet includes long and detailed descriptions, it is not well-suited for evaluating CLIP-based vision-language models like RemoteCLIP (Liu et al., 2024) and RS-CLIP (Li et al., 2023b). Therefore, we focus on evaluating widely used multimodal large language models, including LLaVA-v1.5 (Liu et al., 2023b), MiniGPT-v2 (Chen et al., 2023), MiniGPT-4 (Zhu et al., 2023), and GeoChat (Kuckreja et al., 2023). All experiments are performed using the ChatGPT-4V version of our dataset, which allows us to conduct extensive evaluations across multiple models while significantly reducing computational resource requirements. Note that the prompt used during dataset creation and the instruction prompt for model evaluation are entirely different. Dataset creation involves leveraging land cover maps and designing prompts to generate rich descriptions of satellite images. In contrast, during both training and evaluation of models, only satellite images are used as visual input. To ensure consistency and fairness, all models are evaluated using the same instruction prompt: "Provide a detailed description of the given image" or its variants.

Table 2 summarizes the results of these evaluations, detailing the models' performance across several widely used metrics: BLEU, CIDEr, METEOR, ROUGE-L, and SPICE. We evaluate these models in two experimental settings. The first is a zero-shot transfer setting, where pre-trained models are used to generate captions without any additional training or fine-tuning on the ChatEarthNet dataset. The first four rows in Table 2 present the results of this zero-shot transfer setting. The performance is suboptimal due to the domain gap between the models' original training datasets and our test dataset. Specifically, the original GeoChat model exhibits unsatisfactory zero-shot performance on the ChatEarthNet dataset due to the substantial domain differences between its training datasets and our proposed dataset. GeoChat is trained primarily on high-resolution datasets designed for tasks such as object detection, visual question answering, and scene classification, which lack the global-scale land use and land cover-related semantics and descriptions. The differences in spatial resolution, coupled with the lack of comprehensive land cover content, significantly limit GeoChat's performance on ChatEarthNet. These gaps also motivate the need for ChatEarthNet to complement existing datasets. In addition to zero-shot testing, we fine-tune some of these models on the ChatEarthNet dataset (ChatGPT-4V version) and report their performance. The results clearly show that fine-tuning on our proposed dataset significantly improves image captioning performance in the context of remote sensing data. These findings strongly suggest that ChatEarthNet is a valuable resource for both training and evaluating vision-language geo-foundation models in the remote sensing domain.

**Table 2.** Performance comparison of different models on the ChatEarthNet (ChatGPT-4V Version) test set.

| Models | Bleu-1 | Bleu-2 | Bleu-3 | Bleu-4 | CIDEr | METEOR | ROUGE_L | SPICE |
|---|---|---|---|---|---|---|---|---|
| LLaVA-v1.5 | 0.285 | 0.116 | 0.040 | 0.014 | 0.012 | 0.104 | 0.186 | 0.093 |
| MiniGPT-v2 | 0.279 | 0.116 | 0.041 | 0.015 | 0.009 | 0.104 | 0.180 | 0.091 |
| MiniGPT-4 | 0.175 | 0.072 | 0.023 | 0.008 | 0.000 | 0.116 | 0.180 | 0.079 |
| GeoChat | 0.199 | 0.088 | 0.034 | 0.011 | 0.005 | 0.067 | 0.126 | 0.083 |
| MiniGPT-4 (ChatEarthNet) | 0.310 | 0.184 | 0.113 | 0.071 | 0.001 | **0.209** | 0.254 | 0.186 |
| GeoChat (ChatEarthNet) | **0.445** | **0.269** | **0.170** | **0.109** | **0.094** | 0.208 | **0.286** | **0.211** |

## 4   Code and data availability

The code that utilizes the ChatGPT API to generate captions can be found at https://github.com/zhu-xlab/ChatEarthNet, and its DOI is https://doi.org/10.5281/zenodo.11004358 (Yuan et al., 2024b). The ChatEarthNet dataset is accessible in the Zenodo data repository at https://doi.org/10.5281/zenodo.11003436 (Yuan et al., 2024c). The ChatEarthNet dataset consists of image data and corresponding textual descriptions organized into JSON files. Specifically, there are six JSON files: (1) "ChatEarthNet_caps_35_train.json" and "ChatEarthNet_caps_4v_train.json," which contain image paths and corresponding captions for the training set; (2) "ChatEarthNet_caps_35_val.json" and "ChatEarthNet_caps_4v_val.json," which contain image paths and corresponding captions for the validation set; and (3) "ChatEarthNet_caps_35_test.json" and "ChatEarthNet_caps_4v_test.json," which contain image paths and corresponding captions for the test set. Each JSON file contains a collection of data samples, with each sample comprising an "image_id" field that specifies the image's file path, and a "caption" field that provides a detailed textual description of the corresponding image content. Each Sentinel-2 image in the dataset includes nine spectral bands, which are distributed across three ZIP files. These files are organized as follows: (1) "s2_rgb_images.zip," which contains the RGB bands: Band-R, Band-G, and Band-B; (2) "s2_band_5_6_7_images.zip," which contains the spectral bands: Band-5, Band-6, and Band-7; and (3) "s2_band_8_11_12_images.zip," which contains the spectral bands: Band-8, Band-11, and Band-12.

## 5   Conclusion

In this work, we construct ChatEarthNet, a large-scale image-text dataset characterized by its global coverage, high quality, wide-ranging diversity, and detailed descriptions. Specifically, we utilize Sentinel-2 data for its global coverage as the image source, and we employ land cover maps from ESA's WorldCover project to generate text. Consequently, by analyzing these land cover maps, we manage to extract the spatial distributions of different land cover types, which serve as the context information for crafting the prompts. These well-curated prompts are employed to elicit descriptive captions for Sentinel-2 images from two large language models, ChatGPT-3.5 and ChatGPT-4V. ChatEarthNet comprises 163,488 image-text pairs with captions

generated by ChatGPT-3.5 and an additional 10,000 pairs with captions generated by ChatGPT-4V. By combining high-quality captions with the visual information from Sentinel-2 imagery, ChatEarthNet is a valuable resource for training and evaluating vision-language geo-foundation models for remote sensing. It is worth noting that the proposed ChatEarthNet dataset can be readily used for other tasks, including image-to-text and text-to-image synthesis. Moreover, leveraging the capabilities of large language models, it can also be extended to visual question answering by prompting large language models for questions and answers based on rich descriptions. This versatility enhances the dataset's value to the community.

# Appendix A

We provide image-text pairs from ChatEarthNet to compare the differences between the two versions.

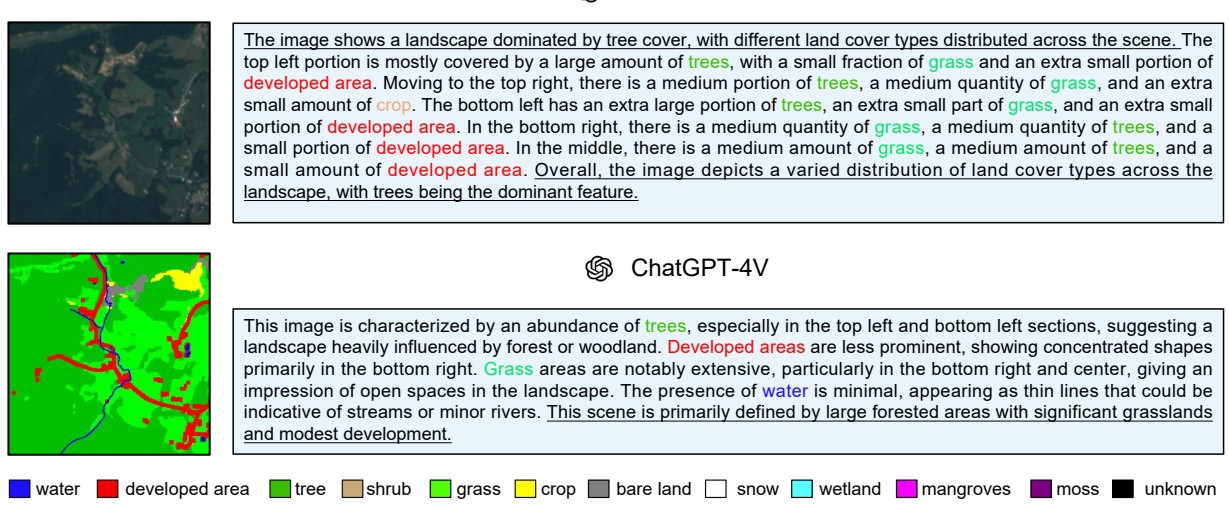

**Figure A1.** Sample 1: Sentinel-2 satellite image, its associated land cover map, and its corresponding captions generated by ChatGPT-3.5 and ChatGPT-4V.

**Figure A2.** Sample 2: Sentinel-2 satellite image, its associated land cover map, and its corresponding captions generated by ChatGPT-3.5 and ChatGPT-4V.

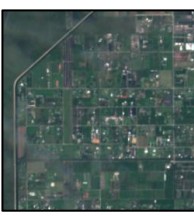

**ChatGPT-3.5**

The image depicts a landscape with various land cover types. In the top left corner, there is a medium amount of wetland, followed by a medium fraction of developed area and a small fraction of grass. Moving to the top right, there is a medium quantity of developed area, followed by a medium amount of grass and a small amount of wetland. In the bottom left, there is a medium amount of developed area, a small portion of grass, and another small portion of crop. As for the bottom right, it contains a medium amount of developed area, a medium part of tree, and a small part of grass. In the middle section, the dominant land cover type is a large fraction of developed area, followed by a medium amount of grass and a small portion of tree. The main theme of the image is the diverse distribution of land cover types in the landscape.

**ChatGPT-4V**

This image presents a diverse and balanced distribution of land covers with a notable percentage of developed areas that exhibit a structured grid-like layout, especially concentrated in the central and bottom right parts of the image. Wetlands are present but much less dominant, scattered throughout the image, with the highest concentration in the top left section. Grasslands appear consistently across the image, whereas the presence of crops is more dispersed. Noticeable areas of trees are found, with the most substantial concentration in the bottom right, providing a contrast to the urban landscape. Water bodies appear in relatively small amounts, and areas with bare soil are interspersed throughout, with shrubland present only in trace amounts.

■ water ■ developed area ■ tree ■ shrub ■ grass ■ crop ■ bare land □ snow ■ wetland ■ mangroves ■ moss ■ unknown

**Figure A3.** Sample 3: Sentinel-2 satellite image, its associated land cover map, and its corresponding captions generated by ChatGPT-3.5 and ChatGPT-4V.

To provide a clear comparison between our dataset and existing ones, we list some representative image-text datasets for remote sensing in the following table.

**Table A1.** A summary of the remote sensing image-text datasets.

| Dataset | #Image-text pairs | Caption Granularity | Caption Generation | Image Data | Geographical Coverage |
|---|---|---|---|---|---|
| UCM-Captions (Qu et al., 2016) | 10,500 | Coarse-grained | Manually Annotated | RGB, UCMerced (Yang and Newsam, 2010) | Regional |
| Sydney-Captions (Qu et al., 2016) | 3,065 | Coarse-grained | Manually Annotated | RGB, Sydney (Zhang et al., 2014) | Regional |
| RSICD (Lu et al., 2017) | 54,605 | Coarse-grained | Manually Annotated | RGB, Google Earth, Baidu Map | Regional |
| NWPU-Captions (Cheng et al., 2022) | 157,500 | Coarse-grained | Manually Annotated | RGB, NWPU-RESISC45 (Cheng et al., 2017) | Regional |
| RSICap (Hu et al., 2023) | 2,585 | Fine-grained | Manually Annotated | RGB, DOTA (Xia et al., 2018) | Regional |
| RS5M (Zhang et al., 2023) | 5,000,000 | Coarse-grained | Model-generated & multiple datasets | RGB, multiple datasets | Global |
| SkyScript (Wang et al., 2024) | 2,600,000 | Coarse-grained | OpenStreetMap | RGB & multispectral, multiple sensors | Global |
| FIT-RS (Luo et al., 2024) | 1,800,851 | Fine-grained | STAR & ChatGPT | RGB, STAR (Li et al., 2024) | Global |
| RemoteCLIP (Liu et al., 2024) | 828,725 | Coarse-grained | Rule-based | RGB, multiple datasets | Global |
| ChatEarthNet | 173,488 | Fine-grained | WorldCover & ChatGPT | RGB&multispectral, Sentinel-2 | Global |

For a clear understanding of the prompts used for generating descriptions, we provide some examples of prompt outputs by Algorithms 1-3.

---

**An example of prompt output by Algorithm 1**

grass; tree; developed area; crop; water; bare land.

The **top left** mainly contains the following land cover types, in descending order of content:

grass (medium part), tree (medium amount), and developed area (medium amount).

The **top right** mainly contains the following land cover types, in descending order of content:

tree (medium quantity), grass (medium amount), and developed area (small amount).

The **bottom left** mainly contains the following land cover types, in descending order of content:

grass (medium amount), tree (medium fraction), and developed area (medium fraction).

The **bottom right** mainly contains the following land cover types, in descending order of content:

crop (medium part), grass (medium portion), and developed area (small portion).

The **middle** mainly contains the following land cover types, in descending order of content:

tree (medium part), grass (medium portion), and developed area (medium fraction).

---

**An example of prompt output by Algorithm 2**

**crop**: top left: 72.27% top right: 43.16% bottom left: 41.78% bottom right: 58.15% middle: 39.85%

**grass**: top left: 10.02% top right: 26.73% bottom left: 36.18% bottom right: 16.22% middle: 27.62%

**developed**: top left: 14.67% top right: 24.27% bottom left: 15.21% bottom right: 21.97% middle: 23.75%

**water**: top left: 1.68% top right: 3.22% bottom left: 2.73% bottom right: 0.21% middle: 3.45%

**bare**: top left: 0.12% top right: 0.21% bottom left: 0.11% bottom right: 0.29% middle: 0.28%

**tree**: top left: 1.04% top right: 1.06% bottom left: 3.98% bottom right: 2.92% middle: 4.46%

---

Note that in Algorithm 2, we calculate the percentage of a specific land cover in each patch, not the percentage of one land cover in the entire image. Therefore, the sum of the percentages is not 1.

---

**An example of prompt output by Algorithm 3**

**top left** distribution: crop: 0.72; developed: 0.15; grass: 0.10; water: 0.02; tree: 0.01; bare: 0.00;

**top right** distribution: crop: 0.43; grass: 0.27; developed: 0.24; water: 0.03; tree: 0.01; bare: 0.00;

**bottom left** distribution: crop: 0.42; grass: 0.36; developed: 0.15; tree: 0.04; water: 0.03; bare: 0.00;

**bottom right** distribution: crop: 0.58; developed: 0.22; grass: 0.16; tree: 0.03; bare: 0.00; water: 0.00;

**middle** distribution: crop: 0.40; grass: 0.28; developed: 0.24; tree: 0.04; water: 0.03; bare: 0.00;

---

**Author contributions**

The dataset was conceptualized by XXZ. The dataset curation and construction were undertaken by ZY, ZX, and LM. The initial manuscript draft was authored by ZY with inputs from all authors. Subsequent revisions of the manuscript were carried out by all authors.

**Competing interests**

The authors declare that they have no conflict of interest.

**Acknowledgements**

This work is jointly supported by the German Federal Ministry of Education and Research (BMBF) in the framework of the international future AI lab "AI4EO – Artificial Intelligence for Earth Observation: Reasoning, Uncertainties, Ethics and Beyond" (grant number: 01DD20001), by the German Federal Ministry for Economic Affairs and Climate Action in the framework of the "national center of excellence ML4Earth" (grant number: 50EE2201C) and by the Munich Center for Machine Learning. We also would like to acknowledge ChatGPT, an AI language model developed by OpenAI.

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
