# Peer review of "ChatEarthNet: A Global-Scale Image-Text Dataset Empowering Vision-Language Geo-Foundation Models"

_Earth System Science Data, 2024_

## Community Comment (CC1)

**essd-2024-140 — Revision**

**ChatEarthNet: A Global-Scale Image-Text Dataset Empowering Vision-Language Geo-Foundation Models**

**By Z. Yuan, Z. Xiong, L. Mou, X. X. Zhu**

**General remarks to all reviewers and editors:**

**We sincerely thank the editors and anonymous reviewers for their valuable comments and suggestions. Below, we provide point-by-point responses to the reviewers' comments. The reviewers' comments are in black, and our responses follow in blue. The revised parts are marked in red in the manuscript.**

**Reviewer #1:**

The authors propose a land cover dataset, ChatEarthNet, built by pairing Sentinel-2 patches with their corresponding WorldCover masks, which contain 12 land cover classes.

The originality comes from providing the land cover data, not directly a a bitmap, but as a textual description extracted from the WorldCover map by means of a large language model (LLM).

Specifically, they use two different models: ChatGPT-3.5, an LLM that can only receive text as input, and ChatGPT-4V, a vision LLM (VLLM) that is able to understand both text and images. Due to cost, they provide 163k images with captions generated by GPT-3.5 and 10k by GPT-4V.

The Sentinel-2 patches are obtained from the dataset SatlasPretrain.

Main comments:

1. The paper describes the prompting process, which differs for GPT-3.5 and -4V. Although the prompt is provided, some details are missing in relation to the exact construction of the outputs of algorithms 1 to 3, since the exact wording of the prompt produced by these algorithms is not given.

**R: We appreciate the reviewer's comment. We have included the exact wording produced by algorithms 1 to 3 in Appendix A. Below are examples of the outputs generated by these algorithms:**

**An example of the exact wording of the prompt generated by Algorithm 1:**

**An example of the exact wording of the prompt generated by Algorithm 2:**

**An example of the exact wording of the prompt generated by Algorithm 3:**

**An example of prompt output by Algorithm 3**

**top left** distribution: crop: 0.72; developed: 0.15; grass: 0.10; water: 0.02; tree: 0.01; bare: 0.00;

**top right** distribution: crop: 0.43; grass: 0.27; developed: 0.24; water: 0.03; tree: 0.01; bare: 0.00;

**bottom left** distribution: crop: 0.42; grass: 0.36; developed: 0.15; tree: 0.04; water: 0.03; bare: 0.00;

**bottom right** distribution: crop: 0.58; developed: 0.22; grass: 0.16; tree: 0.03; bare: 0.00; water: 0.00;

**middle** distribution: crop: 0.40; grass: 0.28; developed: 0.24; tree: 0.04; water: 0.03; bare: 0.00;

**Bold and underline are used to improve readability. Note that in Algorithm 2, we calculate the percentage of a specific land cover in each patch, not the percentage of one land cover in the entire image. Therefore, the sum of the percentages is not 1. These examples are added to Appendix A to provide a clear understanding of how the exact outputs of the algorithms are used to construct the prompts.**

2. Section 2.5 briefly mentions that manual verification is applied in order to check that the LLM correctly followed the prompt instructions. However, it is not clear how many times the prompt had to be modified, and the kind of modifications that were required.

**R: Thanks for the comment. To clarify the potential confusion, the manual verification process described in Section 2.5 aims to ensure the quality and correctness of the generated captions, not modify the prompts. Once the prompts were finalized, we did not further modify them. Instead, our manual verification process focuses on reviewing and correcting the generated captions to ensure they meet our quality standards.**

**Unfortunately, we did not track the number of samples where modifications were made for captions, making it difficult to provide exact answers. However, we would like to emphasize the reason for manual verification: when using ChatGPT-4V, we combine four images into a single API request. We adopt this approach due to the constraints on API usage for ChatGPT-4V (with a tier 1 limit of 500 requests per day by February 2024) and to enhance the efficiency of text generation. Despite providing specific instructions for ChatGPT-4V to treat each image individually, it occasionally makes mistakes by describing comparisons between images, which is not our intention. In such cases, manual corrections are necessary. In contrast, when using the ChatGPT-3.5 model (with a tier 1 limit of 10k requests per day by February 2024), each image is processed through individual API requests.**

3. Although the authors claim that "10k high-quality image-text pairs using ChatGPT-4V are sufficient for fine-tuning large vision-language models", they do not provide any evidence for this. There is not evaluation of the properties of a model trained with the proposed dataset, making it impossible to judge the quality of the representation that can be learned with it, in comparison with a model trained directly for land cover mapping using the WorldCover data.

**R: Thanks for the insightful comments. We appreciate the opportunity to provide further evidence and clarification regarding the quality of our dataset. Our claim that the dataset consists of high-quality image-text pairs is grounded in the following key factors:**

1) **Our dataset provides rich descriptions of the Sentinel-2 images, which contain information about shapes, spatial relationships, distributions, and the main theme of the image, which can be used to train or fine-tune multimodal large language models (MLLMs). Regarding the comparison with traditional segmentation models trained using WorldCover data, a key difference is evident: traditional segmentation models trained on WorldCover data lack the ability to generate rich linguistic descriptions of shapes, spatial relationships, and distributions of land cover types. This limitation indicates the superiority of our dataset in capturing and conveying complex geospatial information through natural language.**

2) **To further support our claim, we provide experiments to fine-tune MLLMs to prove that the proposed dataset can be used to enhance the development of large vision language models. As shown in Table 2, compared with existing MLLMs in the zero-shot setting, like LLaVA-v1.5 [1], MiniGPT-v2 [2], MiniGPT-4 [3], and GeoChat [4], the fine-tuned models using ChatEarthNet (ChatGPT-4V version) can achieve clearly better performance. The results indicate that the proposed ChatEarthNet dataset is**

**not only useful for downstream applications but also effective as a benchmark to evaluate different MLLMs. Please refer to *Comment #4* for more details.**

3) **It is worth noting that the rich descriptions with natural language on Sentinel-2 images in our dataset provide opportunity to non-expert users to understand the sentinel images, who may have difficulty understanding WorldCover labels.**

**Table 2.** Performance comparison of different models on the ChatEarthNet (ChatGPT-4V Version) test set.

| Models | Bleu-1 | Bleu-2 | Bleu-3 | Bleu-4 | CIDEr | METEOR | ROUGE_L | SPICE |
|---|---|---|---|---|---|---|---|---|
| LLaVA-v1.5 | 0.285 | 0.116 | 0.040 | 0.014 | 0.012 | 0.104 | 0.186 | 0.093 |
| MiniGPT-v2 | 0.279 | 0.116 | 0.041 | 0.015 | 0.009 | 0.104 | 0.180 | 0.091 |
| MiniGPT-4 | 0.175 | 0.072 | 0.023 | 0.008 | 0.000 | 0.116 | 0.180 | 0.079 |
| GeoChat | 0.199 | 0.088 | 0.034 | 0.011 | 0.005 | 0.067 | 0.126 | 0.083 |
| MiniGPT-4 (ChatEarthNet) | 0.310 | 0.184 | 0.113 | 0.071 | 0.001 | **0.209** | 0.254 | 0.186 |
| GeoChat (ChatEarthNet) | **0.445** | **0.269** | **0.170** | **0.109** | **0.094** | 0.208 | **0.286** | **0.211** |

4. The authors conclude that "ChatEarthNet is a valuable resource for training and evaluating vision-language geo-foundation models for remote sensing". However, it is not fully clear how this evaluation would work. To be able to conclude this, I suggest the authors do use the dataset to evaluate existing models, such as RemoteCLIP [1], RSCLIP [2] and others.

**R: We thank the reviewer for the valuable suggestion. We agree that demonstrating ChatEarthNet's utility for evaluating vision-language geo-foundation models is crucial. In response to the suggestion, we have conducted benchmarking experiments using various existing models on the proposed dataset.**

**Regarding the reviewer's recommendations to include RemoteCLIP [1] and RSCLIP [2], we appreciate the suggestion. However, the pretrained RSCLIP model is not publicly available at this time. Reproducing its training process would require significant computational resources, which presents substantial challenges. As such, direct evaluation of RSCLIP is currently not feasible.**

**As for RemoteCLIP, while it is a CLIP-based model suitable for vision tasks, applying it directly to ChatEarthNet, which contains long and detailed descriptions, would require extensive alignment with large language models through training connectors on a sizable dataset. This process is resource-intensive and beyond the scope of this paper. We thank the reviewer for pointing out this, and we have added explanations in the revised version, as presented below.**

**To address the reviewer's concerns, we have conducted benchmarking experiments using widely established MLLMs. Specifically, we evaluated several MLLMs, including LLaVA-v1.5 [3], MiniGPT-v2 [4], MiniGPT-4 [5], and GeoChat [6]. These evaluations further support our conclusion that ChatEarthNet is a valuable resource for training and evaluating**

**vision-language geo-foundation models for remote sensing. We have added the experimental part to the revised paper as follows.**

"To demonstrate the effectiveness of ChatEarthNet in evaluating multimodal large language models, we conduct benchmarking experiments using a range of existing models. Given that ChatEarthNet includes long and detailed descriptions, it is not well-suited for evaluating CLIP-based vision-language models like RemoteCLIP [1] and RS-CLIP [2]. Therefore, we focus on evaluating widely used multimodal large language models, including LLaVA-v1.5 [3], MiniGPT-v2 [4], MiniGPT-4 [5], and GeoChat [6]. All experiments are performed using the ChatGPT-4V version of our dataset, which allows us to conduct extensive evaluations across multiple models while significantly reducing computational resource requirements.

Table 2 summarizes the results of these evaluations, detailing the models' performance across several widely used metrics: BLEU, CIDEr, METEOR, ROUGE-L, and SPICE. We evaluate these models in two experimental settings. The first is a zero-shot transfer setting, where pre-trained models are used to generate captions without any additional training or fine-tuning on the ChatEarthNet dataset. The first four rows in Table 2 present the results of this zero-shot transfer setting. The performance is suboptimal due to the domain gap between the models' original training datasets and our test dataset. In addition to zero-shot testing, we fine-tune some of these models on the ChatEarthNet dataset (ChatGPT-4V version) and report their performance. The results clearly show that fine-tuning on our proposed dataset significantly improves image captioning performance in the context of remote sensing data. These findings strongly suggest that ChatEarthNet is a valuable resource for both training and evaluating vision-language geo-foundation models in the remote sensing domain."

**Table 2.** Performance comparison of different models on the ChatEarthNet (ChatGPT-4V Version) test set.

| Models | Bleu-1 | Bleu-2 | Bleu-3 | Bleu-4 | CIDEr | METEOR | ROUGE_L | SPICE |
|---|---|---|---|---|---|---|---|---|
| LLaVA-v1.5 | 0.285 | 0.116 | 0.040 | 0.014 | 0.012 | 0.104 | 0.186 | 0.093 |
| MiniGPT-v2 | 0.279 | 0.116 | 0.041 | 0.015 | 0.009 | 0.104 | 0.180 | 0.091 |
| MiniGPT-4 | 0.175 | 0.072 | 0.023 | 0.008 | 0.000 | 0.116 | 0.180 | 0.079 |
| GeoChat | 0.199 | 0.088 | 0.034 | 0.011 | 0.005 | 0.067 | 0.126 | 0.083 |
| MiniGPT-4 (ChatEarthNet) | 0.310 | 0.184 | 0.113 | 0.071 | 0.001 | **0.209** | 0.254 | 0.186 |
| GeoChat (ChatEarthNet) | **0.445** | **0.269** | **0.170** | **0.109** | **0.094** | 0.208 | **0.286** | **0.211** |

[1] F. Liu, D. Chen, Z. Guan, X. Zhou, J. Zhu, Q. Ye, L. Fu, and J. Zhou, "RemoteCLIP: A vision language foundation model for remote sensing," *IEEE Transactions on Geoscience and Remote Sensing*, 2024.
[2] X. Li, C. Wen, Y. Hu, and N. Zhou, "RS-CLIP: Zero shot remote sensing scene classification via contrastive vision-language supervision," *International Journal of Applied Earth Observation and Geoinformation*, 2023.
[3] H. Liu, C. Li, Q. Wu, and Y. J. Lee, "Visual instruction tuning," *Advances in Neural Information Processing Systems 36*, 2023.

[4] J. Chen, D. Zhu, X. Shen, X. Li, Z. Liu, P. Zhang, R. Krishnamoorthi, V. Chandra, Y. Xiong, and M. Elhoseiny, "MiniGPT-v2: Large language model as a unified interface for vision-language multi-task learning," *arXiv preprint arXiv:2310.09478*, 2023.

[5] D. Zhu, J. Chen, X. Shen, X. Li, and M. Elhoseiny, "MiniGPT-4: Enhancing vision-language understanding with advanced large language models," *arXiv preprint arXiv:2304.10592*, 2023.

[6] K. Kuckreja, M. S. Danish, M. Naseer, A. Das, S. Khan, and F. S. Khan, "Geochat: Grounded large vision-language model for remote sensing," *Proceedings of the IEEE/CVF Conference on Computer Vision and Pattern Recognition*, 2024.

Minor comments:

1. In Section 3.2, the authors write that "ChatGPT-3.5 is more dense, covering a wider range of areas". However, aren't both datasets obtained by randomly sampling SatlasPretain? Shouldn't they therefore have roughly the same distribution? If I understand it well, the only difference should be the number of images.

R: Thank you for your question. You are correct. Due to the cost and access limitations of ChatGPT-4V, the number of images used in ChatGPT-4V is significantly lower compared to the number used in ChatGPT-3.5. Regarding the geographical coverage, they basically have roughly the same distribution. The only difference is the density of coverage.

2. In Section 3.3, they authors explore word frequency in the generated captions.

R: The word frequency analysis in Section 3.3 provides valuable insights into the linguistic characteristics of the generated captions.

3. In line 50, "few pairs in the website" should be "few pairs on the web" or "online".

R: Thank the reviewer for pointing out this. We have revised the relevant sentence to: "However, few pairs on the web provide detailed descriptions for satellite images." Please kindly check out the revised version.

4. The authors often refer to "land covers", although may be "land cover types" or "classses" would be more appropriate.

R: We appreciate the reviewer's suggestions on this point. We agree that "land cover types" or "classes" are more appropriate terms. We have revised the relevant terms in the manuscript.

1. Throughout the paper, the author mentioned image-text datasets many times. Image-text datasets cover multiple different types of annotations, such as image caption, VQA, and visual grounding. Since this paper focuses on image captioning, the writing should be modified accordingly.

**R: We appreciate the reviewer's suggestion on this point. We agree with the reviewer's understanding of the concept of image-text dataset. Image-text datasets indeed contain multiple types of text annotations, including image captioning, visual question answering, and visual grounding. However, our dataset specifically provides long, detailed descriptions of images, particularly focusing on land cover types and their spatial distribution. While these descriptions are closely related to image captioning tasks, they contain richer information that extends beyond typical captioning tasks. It can be readily used for other generative tasks, such as image-to-text and text-to-image synthesis. In addition, the dataset can also be easily extended to visual question answering by leveraging the capabilities of current large language models. This versatility is why we refer to it as an "image-text dataset," a high-level term that captures its potential for a range of tasks.**

**In light of this, we choose to use the broader term "image-text dataset" to reflect the higher-level concept of images paired with textual descriptions. This is consistent with prior works [1]-[5], which also use "image-text dataset" when focusing primarily on image captioning tasks.**

**[1] D. Qi, L. Su, J. Song, E. Cui, T. Bharti, and A. Sacheti, "ImageBERT: Cross-modal pre-training with large-scale weak-supervised image-text data," *arXiv preprint arXiv:2001.07966*, 2020.**
**[2] K. Srinivasan, K. Raman, J. Chen, M. Bendersky, and M. Najork, "WIT: Wikipedia-based image text dataset for multimodal multilingual machine learning," *Proceedings of the 44th International ACM SIGIR Conference on Research and Development in Information Retrieval*, 2021.**
**[3] K. Desai, G. Kaul, Z. Aysola, and J. Johnson, "RedCaps: Web-curated image-text data created by the people, for the people", *arXiv preprint arXiv:2111.11431*, 2021.**
**[4] Y. Okamoto, H. Toyonaga, Y. Ijiri, and H. Kataoka, "Constructing image-text pair dataset from books," *arXiv preprint arXiv:2310.01936*, 2023.**
**[5] Q. Yu, Q. Sun, X. Zhang, Y. Cui, F. Zhang, Y. Cao, X. Wang, and J. Liu, "Capsfusion: Rethinking image-text data at scale," *Proceedings of the IEEE/CVF Conference on Computer Vision and Pattern Recognition*, 2024.**

2. The authors use land cover labels from WoldCover products to formulate prompts. The information carried by image captions mainly covers land cover information, this limits the usage of the proposed dataset. This is a big drawback when compared to previous datasets (e.g., RSICap) that provide more diverse information (such as object counting, position, size, and complex reasoning).

**R: Thanks for the insightful comments. RSICap [6] is an excellent dataset that offers diverse and detailed annotations, but it has a relatively smaller volume and relies on manual**

**annotation. In contrast, our dataset leverages automated methods to generate a significantly larger volume of image-text pairs, ensuring broader coverage and scalability. Our dataset consists of satellite images with global coverage and lower resolution. This makes object counting and complex reasoning more challenging due to the granularity of the images.**

**Although we use land cover labels from WorldCover products to formulate prompts, our dataset also includes detailed descriptions related to position and size. For example: "This image reveals a mix of developed areas and trees, with developed areas showing expansive coverage particularly in the top left, signifying widespread human settlement or infrastructure. Bodies of water are substantially present, especially in the top left, forming large open shapes indicative of lakes or wide rivers. Trees spread significantly across the bottom half, offering a sense of a forested or natural region, while grasslands are present but less dominant. Varying shapes in the pattern of developed areas and the strong presence of water features characterize this image alongside the notable forest coverage."**

**In summary, we believe that both RSICap and our dataset offer valuable contributions to the community, but with distinct focuses. RSICap emphasizes high-resolution object recognition, counting, and attribute analysis, while ChatEarthNet focuses on land cover types and global coverage.**

**[6] Y. Hu, J. Yuan, C. Wen, X. Lu, and X. Li, "RSGPT: A remote sensing vision language model and benchmark." *arXiv preprint arXiv:2307.15266*, 2023.**

3. Information Overlap. To generate image captions, the proposed method divides each image of 256x256 into 5 patches of size 128x128, top-left, top-right, bottom-left, bottom-right, and middle patches. The center patch overlaps with other patches. This causes two issues: 1) duplicated object description; 2) duplicated object counting.

**R: We appreciate the reviewer's comment on this point. In real-world scenarios, it is common for land cover types to span across multiple regions, including overlapping areas. While our method involves dividing the image into patches with some overlap, this does not affect describing each patch individually. Specifically, we divide one image into five patches, including a central one that overlaps with the others, to ensure a comprehensive description of the spatial distribution across the entire image. Without this overlap, the central portion of the image might be overlooked, leading to incomplete coverage of the spatial pattern.**

**Since our dataset focuses on land cover types, which often lack distinct object boundaries, there is no issue of duplicated object descriptions. We also notice that there are no redundant descriptions in overlapping areas. Additionally, ChatEarthNet does not involve object counting in its captions. Therefore, the overlap does not introduce any issues related to duplicated object counting, ensuring that the dataset remains unaffected in this case.**

4. "Moreover, considering the API request limit of ChatGPT-4V, we put four images into one request to generate descriptions more efficiently". By putting four images into one request, do you mean concate the images into one? Merging multiple images will cause undesired interactions between image features caused by self-attention in transformer architecture. As far as I know,

GPT-4V allows 10,000 requests per day, it's therefore not necessary to put four images into one request.

**R: Thank you for the insightful comment. To clarify, we do not concatenate four images into one. Instead, we send four separate images in a single API request to ChatGPT-4V. Therefore, there are no interactions between image features at the model level. However, in a few cases, the returned descriptions include comparisons between different images, which is not our intention. To address this, we manually review and correct such descriptions to ensure quality.**

**In addition, we would like to clarify three points regarding our decision to put four images into a single request when generating captions with ChatGPT-4V.**

1) **Our work began in 2023, and the first version of the manuscript was submitted in February 2024. At that time, for usage tier 1, the limit was set at 500 requests per day, not the 10,000 requests per day that are available now. Given the resource constraints we faced at that time, we chose to put four images into one request to generate descriptions more efficiently.**
2) **To ensure that the images are described independently, our prompts specifically request: "Generate the four descriptions separately; do not add connections between them."**
3) **Despite the prompt requesting no interactions between images, some descriptions still contain comparisons among the four images. To ensure quality, we manually check all captions generated by ChatGPT-4V and refine comparison-related captions.**

5. Missing experimental verification. By the current version, it's unclear how this dataset can be used to boost the development of LVLMs in remote sensing. As a benchmark dataset, it's better to show the image captioning performance of existing well-known methods on the proposed dataset.

**R: We thank the reviewer for the valuable comment. We agree that demonstrating ChatEarthNet's utility for evaluating vision-language geo-foundation models is crucial. To address the reviewer's concerns, we have conducted additional benchmarking experiments using widely established multimodal large language models (MLLMs). Specifically, we evaluated several MLLMs, including LLaVA-v1.5 [7], MiniGPT-v2 [8], MiniGPT-4 [9], and GeoChat [10]. These evaluations further support our conclusion that ChatEarthNet is a valuable resource for training and evaluating vision-language geo-foundation models for remote sensing. Please kindly check out the revised version as follows.**

"To demonstrate the effectiveness of ChatEarthNet in evaluating multimodal large language models, we conduct benchmarking experiments using a range of existing models. Given that ChatEarthNet includes long and detailed descriptions, it is not well-suited for evaluating CLIP-based vision-language models like RemoteCLIP [11] and RS-CLIP [12]. Therefore, we focus on evaluating widely used multimodal large language models, including LLaVA-v1.5 [7], MiniGPT-v2 [8], MiniGPT-4 [9], and GeoChat [10]. All experiments are performed using the ChatGPT-4V version of our dataset, which allows us to conduct extensive evaluations across multiple models while significantly reducing computational resource requirements.

Table 2 summarizes the results of these evaluations, detailing the models' performance across several widely used metrics: BLEU, CIDEr, METEOR, ROUGE-L, and SPICE. We evaluate these models in two experimental settings. The first is a zero-shot transfer setting, where pre-trained models are used to generate captions without any additional training or fine-tuning on the ChatEarthNet dataset. The first four rows in Table 2 present the results of this zero-shot transfer setting. The performance is suboptimal due to the domain gap between the models' original training datasets and our test dataset. In addition to zero-shot testing, we fine-tune some of these models on the ChatEarthNet dataset (ChatGPT-4V version) and report their performance. The results clearly show that fine-tuning on our proposed dataset significantly improves image captioning performance in the context of remote sensing data. These findings strongly suggest that ChatEarthNet is a valuable resource for both training and evaluating vision-language geo-foundation models in the remote sensing domain."

**Table 2.** Performance comparison of different models on the ChatEarthNet (ChatGPT-4V Version) test set.

| Models | Bleu-1 | Bleu-2 | Bleu-3 | Bleu-4 | CIDEr | METEOR | ROUGE_L | SPICE |
|---|---|---|---|---|---|---|---|---|
| LLaVA-v1.5 | 0.285 | 0.116 | 0.040 | 0.014 | 0.012 | 0.104 | 0.186 | 0.093 |
| MiniGPT-v2 | 0.279 | 0.116 | 0.041 | 0.015 | 0.009 | 0.104 | 0.180 | 0.091 |
| MiniGPT-4 | 0.175 | 0.072 | 0.023 | 0.008 | 0.000 | 0.116 | 0.180 | 0.079 |
| GeoChat | 0.199 | 0.088 | 0.034 | 0.011 | 0.005 | 0.067 | 0.126 | 0.083 |
| MiniGPT-4 (ChatEarthNet) | 0.310 | 0.184 | 0.113 | 0.071 | 0.001 | **0.209** | 0.254 | 0.186 |
| GeoChat (ChatEarthNet) | **0.445** | **0.269** | **0.170** | **0.109** | **0.094** | 0.208 | **0.286** | **0.211** |

[7] H. Liu, C. Li, Q. Wu, and Y. J. Lee, "Visual instruction tuning," *Advances in Neural Information Processing Systems 36*, 2023.

[8] J. Chen, D. Zhu, X. Shen, X. Li, Z. Liu, P. Zhang, R. Krishnamoorthi, V. Chandra, Y. Xiong, and M. Elhoseiny, "MiniGPT-v2: Large language model as a unified interface for vision-language multi-task learning," *arXiv preprint arXiv:2310.09478*, 2023.

[9] D. Zhu, J. Chen, X. Shen, X. Li, and M. Elhoseiny, "MiniGPT-4: Enhancing vision-language understanding with advanced large language models," *arXiv preprint arXiv:2304.10592*, 2023.

[10] K. Kuckreja, M. S. Danish, M. Naseer, A. Das, S. Khan, and F. S. Khan, "Geochat: Grounded large vision-language model for remote sensing," *Proceedings of the IEEE/CVF Conference on Computer Vision and Pattern Recognition*, 2024.

[11] F. Liu, D. Chen, Z. Guan, X. Zhou, J. Zhu, Q. Ye, L. Fu, and J. Zhou, "RemoteCLIP: A vision language foundation model for remote sensing," *IEEE Transactions on Geoscience and Remote Sensing*, 2024.

[12] X. Li, C. Wen, Y. Hu, and N. Zhou, "RS-CLIP: Zero shot remote sensing scene classification via contrastive vision-language supervision," *International Journal of Applied Earth Observation and Geoinformation*, 2023.

Minors:

1. ChatGPT-3.5 is not a widely used term. Instead, ChatGPT and gpt-3.5-turbo are more frequently used.

**R: Thanks for the valuable comment. In this manuscript, we use "ChatGPT-3.5" to refer to the model technically known as "gpt-3.5-turbo." Similarly, "ChatGPT-4V" refers to "gpt-4-vision-preview." These terms are intended to provide a more intuitive understanding of the models' positions within the ChatGPT series.**

**We have added the following sentence in the revised manuscript to clarify this as follows.**

"In this manuscript, ChatGPT-3.5 refers to the model gpt-3.5-turbo and ChatGPT-4V refers to the model gpt-4-vision-preview."

2. In line 67, referring image segmentation belongs to visual grounding and therefore should be merged.

**R: Thank you for your comment. We believe you may be referring to line 37. While both visual grounding and referring image segmentation are vision-language tasks, they produce different types of outputs. Visual grounding generates a bounding box around the referred object, while referring image segmentation produces a pixel-level mask for the object based on the query. Given this fundamental difference in output, we choose to keep them as separate tasks in the manuscript.**

3. In line 44, when mentioning large vision-language foundation models, the authors fail to cover popular models, such as MiniGPT-4, and QWen-VL.

**R: Thank you for pointing out this oversight. We have revised the manuscript to include more models as follows.**

"For large vision-language foundation models, CLIP [13], LLaVA [7], MiniGPT-4 [9], MiniGPT-v2 [8], and Qwen-VL [14] have revolutionized the computer vision community."

**[7] H. Liu, C. Li, Q. Wu, and Y. J. Lee, "Visual instruction tuning,"** *Advances in Neural Information Processing Systems 36,* **2023.**

**[8] J. Chen, D. Zhu, X. Shen, X. Li, Z. Liu, P. Zhang, R. Krishnamoorthi, V. Chandra, Y. Xiong, and M. Elhoseiny, "MiniGPT-v2: Large language model as a unified interface for vision-language multi-task learning,"** *arXiv preprint arXiv:2310.09478,* **2023.**

**[9] D. Zhu, J. Chen, X. Shen, X. Li, and M. Elhoseiny, "MiniGPT-4: Enhancing vision-language understanding with advanced large language models,"** *arXiv preprint arXiv:2304.10592,* **2023.**

**[13] A. Radford, J. W. Kim, C. Hallacy, A. Ramesh, G. Goh, S. Agarwal, G. Sastry, A. Askell, P. Mishkin, J. Clark, G. Krueger, and I. Sutskever, "Learning transferable visual models from natural language supervision,"** *Proceedings of the 38th International Conference on Machine Learning,* **2021.**

**[14] J. Bai, S. Bai, S. Yang, S. Wang, S. Tan, P. Wang, J. Lin, C. Zhou, and J. Zhou, "Qwen-VL: A Versatile Vision-Language Model for Understanding, Localization, Text Reading, and Beyond,"** *arXiv preprint arXiv:2308.12966,* **2023.**

4. In Table I, it's unclear whether the 10,000 images used with GPT-4V are included in those 163,488 images used with GPT-3.5. If included, the second column can be removed.

**R: Thanks for the insightful comment. The 10,000 images used with ChatGPT-4V are included in those 163,488 images used with GPT-3.5. Following the reviewer's suggestion, we have removed the second column in Table I in the revised manuscript.**

**Table 1.** The number of Sentinel-2 images used for generating captions, along with the corresponding numbers of captions generated by ChatGPT-3.5 and ChatGPT-4V.

| Subsets | Number of ChatGPT-3.5 Captions | Number of ChatGPT4-V Captions |
|---------|-------------------------------|-------------------------------|
| Train | 98,092 | 6000 |
| Val | 16,348 | 1000 |
| Test | 49,048 | 3000 |
| Sum | 163,488 | 10,000 |

5. In Fig. 15, it's better to show the y-axis with probability distribution instead of No. images for a fair comparison between GPT-3.5 and GPT-4.

**R: Thanks for the valuable comment. We have revised the Fig. 15 to normalize the frequency for a better visual comparison. Please kindly check it out as follows.**

[Figure]

6. Section 3.3 can be compressed.

**R: Thank you for the insightful suggestion. We have revised Section 3.3 to make it more concise. Please kindly check out the revised manuscript.**

The authors propose an image-text dataset for remote sensing vision-language geo-foundation models. In detail, the image source is from Sentinel-2 data, and the descriptions of land covers is obtained from the semantic segmentation labels of the European Space Agency's WorldCover project. Moreover, ChatGPT and the manual verification process are introduced to enhance the dataset. The presented work focus on considerable data collection and processing, however the experimentation could be further improved. The reviewer has the following comments:

Main comments:

1. In this work, a global image-text dataset is presented in the field of remote sensing. There are some existing image-text datasets, and the authors are encouraged to specifically compare the proposed dataset with those that exist, such as SkySenseGPT (https://arxiv.org/pdf/2406.10100), SkyScript (https://ojs.aaai.org/index.php/AAAI/article/view/28393), and RemoteCLIP (https://ieeexplore.ieee.org/document/10504785).

**R: Thanks for this valuable suggestion. We have added comparisons with the mentioned datasets in Appendix A. Please kindly check out the table as follows.**

**Table A1.** A summary of the remote sensing image-text datasets.

| Dataset | #Image-text pairs | Caption Granularity | Caption Generation | Image Data | Geographical Coverage |
|---|---|---|---|---|---|
| UCM-Captions (Qu et al., 2016) | 10,500 | Coarse-grained | Manually Annotated | RGB, UCMerced (Yang and Newsam, 2010) | Regional |
| Sydney-Captions (Qu et al., 2016) | 3,065 | Coarse-grained | Manually Annotated | RGB, Sydney (Zhang et al., 2014) | Regional |
| RSICD (Lu et al., 2017) | 54,605 | Coarse-grained | Manually Annotated | RGB, Google Earth, Baidu Map | Regional |
| NWPU-Captions (Cheng et al., 2022) | 157,500 | Coarse-grained | Manually Annotated | RGB, NWPU-RESISC45 (Cheng et al., 2017) | Regional |
| RSICap (Hu et al., 2023) | 2,585 | Fine-grained | Manually Annotated | RGB, DOTA (Xia et al., 2018) | Regional |
| RS5M (Zhang et al., 2023) | 5 M | Coarse-grained | Model-generated & multiple datasets | RGB, multiple datasets | Global |
| SkyScript (Wang et al., 2024) | 2.6 M | Coarse-grained | OpenStreetMap | RGB & multispectral, multiple sensors | Global |
| FIT-RS (Luo et al., 2024) | 1,800,851 | Fine-grained | STAR & ChatGPT | RGB, STAR (Li et al., 2024) | Global |
| RemoteCLIP (Liu et al., 2024) | 828,725 | Coarse-grained | Rule-based | RGB, multiple datasets | Global |
| ChatEarthNet | 173,488 | Fine-grained | WorldCover & ChatGPT | RGB&multispectral, Sentinel-2 | Global |

**Although the number of FIT-RS dataset proposed in the SkySenseGPT paper is greater than that in ChatEarthNet, this work was submitted to arXiv in June 2024, which is four months later than the submission of ChatEarthNet to arXiv in February 2024.**

2. For the designed dataset, how to consider the imbalance between foreground and background in the remote sensing segmentation task?

**R: Thank you for your question. In our dataset, we use land cover maps to generate detailed descriptions of all land cover types present in the images. As a result, there is no explicit "background" in the sense. Each region in the image is represented by a specific land cover type, as shown in Figs A1, A2, and A3 in Appendix A. Consequently, there is no imbalance issue between foreground and background, as all land cover types are treated equally in the descriptions.**

3. The authors are advised to explain the reasons for choosing the land cover maps from WorldCover. In addition, how to measure the accuracy of labelling in these land cover maps?

**R: Thanks for the valuable comment. WorldCover is selected for our dataset due to its high accuracy and comprehensive land cover types when compared to other available products. How to measure the accuracy of the global land cover products is challenging. To address this issue, Xu et al. conducted a comparative independent validation of recent 10m global land cover maps. As demonstrated in the study by Xu et al. [1], WorldCover outperforms other alternatives such as Dynamic World [2] and ESRI LULC [3], offering more accurate labels and more land cover types. These factors inspire us to choose WorldCover for constructing our dataset instead of others.**

**[1] P. Xu, N. E. Tsendbazar, M. Herold, S. de Bruin, M. Koopmans, T. Birch, S. Carter, S. Fritz, M. Lesiv, E. Mazur, A. Pickens, P. Potapov, F. Stolle, A. Tyukavina, R. Van De Kerchove, and D. Zanaga, "Comparative validation of recent 10 m-resolution global land cover maps," *Remote Sensing of Environment*, 2024.**
**[2] C. F. Brown, S. P. Brumby, B. Guzder-Williams, T. Birch, S. B. Hyde, J. Mazzariello, W. Czerwinski, V. J. Pasquarella, R. Haertel, S. Ilyushchenko, K. Schwehr, M. Weisse, F. Stolle, C. Hanson, O. Guinan, R. Moore, and A. M. Tait, "Dynamic World, near real-time global 10 m land use land cover mapping," *Scientific Data*, 2022.**
**[3] K. Karra, C. Kontgis, Z. Statman-Weil, J. C. Mazzariello, M. Mathis, and S. P. Brumby, "Global land use / land cover with Sentinel 2 and deep learning," *IEEE International Geoscience and Remote Sensing Symposium*, 2021.**

4. The authors mentioned that the proposed dataset has many high-quality and detailed descriptions, and is it validated by quantitative comparison experiments with other datasets?

**R: We appreciate the reviewer's comment on this point. We provided a table to compare the proposed dataset with existing ones as follows.**

**Table A1.** A summary of the remote sensing image-text datasets.

| Dataset | #Image-text pairs | Caption Granularity | Caption Generation | Image Data | Geographical Coverage |
|---|---|---|---|---|---|
| UCM-Captions (Qu et al., 2016) | 10,500 | Coarse-grained | Manually Annotated | RGB, UCMerced (Yang and Newsam, 2010) | Regional |
| Sydney-Captions (Qu et al., 2016) | 3,065 | Coarse-grained | Manually Annotated | RGB, Sydney (Zhang et al., 2014) | Regional |
| RSICD (Lu et al., 2017) | 54,605 | Coarse-grained | Manually Annotated | RGB, Google Earth, Baidu Map | Regional |
| NWPU-Captions (Cheng et al., 2022) | 157,500 | Coarse-grained | Manually Annotated | RGB, NWPU-RESISC45 (Cheng et al., 2017) | Regional |
| RSICap (Hu et al., 2023) | 2,585 | Fine-grained | Manually Annotated | RGB, DOTA (Xia et al., 2018) | Regional |
| RS5M (Zhang et al., 2023) | 5 M | Coarse-grained | Model-generated & multiple datasets | RGB, multiple datasets | Global |
| SkyScript (Wang et al., 2024) | 2.6 M | Coarse-grained | OpenStreetMap | RGB & multispectral, multiple sensors | Global |
| FIT-RS (Luo et al., 2024) | 1,800,851 | Fine-grained | STAR & ChatGPT | RGB, STAR (Li et al., 2024) | Global |
| RemoteCLIP (Liu et al., 2024) | 828,725 | Coarse-grained | Rule-based | RGB, multiple datasets | Global |
| ChatEarthNet | 173,488 | Fine-grained | WorldCover & ChatGPT | RGB&multispectral, Sentinel-2 | Global |

**We also conduct experiments to evaluate widely established multimodal large language models (MLLMs). Specifically, we evaluate several MLLMs, including LLaVA [4], MiniGPT-v2 [5], MiniGPT-4 [6], and GeoChat [7]. These evaluations further support our conclusion that ChatEarthNet is a valuable resource for training and evaluating vision-language geo-foundation models for remote sensing. Please kindly check out the revised version as follows.**

"To demonstrate the effectiveness of ChatEarthNet in evaluating multimodal large language models, we conduct benchmarking experiments using a range of existing models. Given that ChatEarthNet includes long and detailed descriptions, it is not well-suited for evaluating CLIP-based vision-language models like RemoteCLIP [8] and RS-CLIP [9]. Therefore, we focus on evaluating widely used multimodal large language models, including LLaVA-v1.5 [4], MiniGPT-v2 [5], MiniGPT-4 [6], and GeoChat [7]. All experiments are performed using the ChatGPT-4V version of our dataset, which allows us to conduct extensive evaluations across multiple models while significantly reducing computational resource requirements.

Table 2 summarizes the results of these evaluations, detailing the models' performance across several widely used metrics: BLEU, CIDEr, METEOR, ROUGE-L, and SPICE. We evaluate these models in two experimental settings. The first is a zero-shot transfer setting, where pre-trained models are used to generate captions without any additional training or fine-tuning on the ChatEarthNet dataset. The first four rows in Table 2 present the results of this zero-shot transfer setting. The performance is suboptimal due to the domain gap between the models' original training datasets and our test dataset. In addition to zero-shot testing, we fine-tune some of these models on the ChatEarthNet dataset (ChatGPT-4V version) and report their performance. The results clearly show that fine-tuning on our proposed dataset significantly improves image captioning performance in the context of remote sensing data. These findings strongly suggest that ChatEarthNet is a valuable resource for both training and evaluating vision-language geo-foundation models in the remote sensing domain."

**Table 2.** Performance comparison of different models on the ChatEarthNet (ChatGPT-4V Version) test set.

| Models | Bleu-1 | Bleu-2 | Bleu-3 | Bleu-4 | CIDEr | METEOR | ROUGE_L | SPICE |
|---|---|---|---|---|---|---|---|---|
| LLaVA-v1.5 | 0.285 | 0.116 | 0.040 | 0.014 | 0.012 | 0.104 | 0.186 | 0.093 |
| MiniGPT-v2 | 0.279 | 0.116 | 0.041 | 0.015 | 0.009 | 0.104 | 0.180 | 0.091 |
| MiniGPT-4 | 0.175 | 0.072 | 0.023 | 0.008 | 0.000 | 0.116 | 0.180 | 0.079 |
| GeoChat | 0.199 | 0.088 | 0.034 | 0.011 | 0.005 | 0.067 | 0.126 | 0.083 |
| MiniGPT-4 (ChatEarthNet) | 0.310 | 0.184 | 0.113 | 0.071 | 0.001 | **0.209** | 0.254 | 0.186 |
| GeoChat (ChatEarthNet) | **0.445** | **0.269** | **0.170** | **0.109** | **0.094** | 0.208 | **0.286** | **0.211** |

**[4] H. Liu, C. Li, Q. Wu, and Y. J. Lee, "Visual instruction tuning,"** *Advances in Neural Information Processing Systems 36,* **2023.**
**[5] J. Chen, D. Zhu, X. Shen, X. Li, Z. Liu, P. Zhang, R. Krishnamoorthi, V. Chandra, Y. Xiong, and M. Elhoseiny, "MiniGPT-v2: Large language model as a unified interface for vision-language multi-task learning,"** *arXiv preprint arXiv:2310.09478,* **2023.**

[6] D. Zhu, J. Chen, X. Shen, X. Li, and M. Elhoseiny, "MiniGPT-4: Enhancing vision-language understanding with advanced large language models," *arXiv preprint arXiv:2304.10592*, 2023.

[7] K. Kuckreja, M. S. Danish, M. Naseer, A. Das, S. Khan, and F. S. Khan, "Geochat: Grounded large vision-language model for remote sensing," *Proceedings of the IEEE/CVF Conference on Computer Vision and Pattern Recognition*, 2024.

[8] F. Liu, D. Chen, Z. Guan, X. Zhou, J. Zhu, Q. Ye, L. Fu, and J. Zhou, "RemoteCLIP: A vision language foundation model for remote sensing," *IEEE Transactions on Geoscience and Remote Sensing*, 2024.

[9] X. Li, C. Wen, Y. Hu, and N. Zhou, "RS-CLIP: Zero shot remote sensing scene classification via contrastive vision-language supervision," *International Journal of Applied Earth Observation and Geoinformation*, 2023.

5. The authors are encouraged to discuss which attributes are more important for the multimodal vision-language learning than for the vision representation, e.g., relative size or relative position described in the text.

R: Thanks for the insightful comment. In multimodal vision-language learning, attributes like relative size and relative position are important because they provide contextual information that bridges the gap between visual data and natural language. While vision-only representations are good at capturing visual features such as color, texture, and shape, they may fall short in conveying spatial relationships and comparative attributes inherent in complex scenes like satellite imagery. For instance, understanding that "a small lake is nestled beside a large forest" requires integrating both visual cues and linguistic descriptions to fully comprehend the scene.

ChatEarthNet emphasizes these attributes in the generated descriptions. By employing detailed prompts and leveraging semantic segmentation labels from the WorldCover project, we ensure that the natural language descriptions include rich details about relative sizes and positions. This enriches the dataset, making it more suitable for training models that need to understand and generate descriptions involving spatial relationships and comparative sizes.

We believe that highlighting these attributes enhances the performance of multimodal models in tasks such as image captioning, scene understanding, and geospatial analysis. It allows models to develop a more comprehensive understanding of the scene by aligning visual features with corresponding textual descriptions that capture both absolute and relative attributes. In contrast to vision-only models, which might detect objects without understanding their spatial relationships, multimodal models can interpret and describe how different elements in an image relate to one another, leading to more informative and accurate outputs.

We appreciate the reviewer's insightful comment. However, this manuscript mainly focuses on the construction and analysis of the dataset. In the future work, we plan to conduct experiments to quantify the impact of these attributes on model performance.

6. It seems that there is a lot of textual information described in the proposed dataset, does this introduce interfering information? How to avoid negative learning due to interfering information?

**R: Thank you for your insightful comment. We acknowledge that incorporating extensive textual information can introduce challenges, such as noise or irrelevant details that might negatively impact model training. However, with careful prompt design and semantic guidance, we can mitigate these concerns, ensuring that the dataset enhances learning rather than hinders it.**

**We utilize detailed and carefully designed prompts to guide ChatGPT in generating descriptions that are both informative and relevant, avoiding inaccuracies, redundancies, or irrelevant details that could introduce noise. For example, we only focus on three main land cover types in Algorithm 1 instead of all land cover types. Moreover, by incorporating semantic segmentation labels from the WorldCover project, we ensure that the descriptions focus on land cover types and spatial relationships in the image. This semantic guidance helps filter out irrelevant information and emphasizes attributes that are crucial for understanding and interpreting remote sensing data.**

**From the perspective of dataset construction, we have implemented several strategies to enhance quality. However, mitigating negative learning mainly depends on model design, which is beyond the scope of this paper focusing on introducing the dataset. Nevertheless, we believe this is a valuable research direction and intend to pursue it in our future work.**

Minors

1. Please rephrase the description of "Image-Text Dataset", could the proposed dataset be used with other vision-language tasks, such as, image-to-text and text-to-image synthesis?

**R: Thank you for your valuable comment and question. Yes, the ChatEarthNet dataset can indeed be readily used for other generative tasks, such as image-to-text and text-to-image synthesis. In addition, the dataset can also be easily extended to visual question answering by leveraging the capabilities of current large language models.**

**This versatility is why we refer to it as an "image-text dataset," a high-level term that captures its potential for a range of tasks. We have added further clarification in the revised manuscript as follows:**

"It is worth noting that the proposed ChatEarthNet dataset can be readily used for other tasks, including image-to-text and text-to-image synthesis. Moreover, leveraging the capabilities of large language models, it can also be extended to visual question answering by prompting large language models for questions and answers based on rich descriptions. This versatility enhances the dataset's value to the community."

2. The "2.5 Manual verification" section is suggested to add details of manual adjustments, such as under what circumstances manual verification are required and what information is adjusted. An example is visual representation.

**R: Thank you for your valuable suggestion. We have added more details to section "2.5 Manual verification" to better present the manual adjustment process. Please kindly check out the revised version as follows.**

"To avoid unexpected descriptions on comparisons between different images, we design prompts like "Generate the four descriptions separately; do not add connections between them" to guide the description generation process. Despite providing specific instructions for ChatGPT-4V to treat each image individually, it occasionally make mistakes by describing comparisons between images. For instance, phrases such as "similar to other images" and "compared with previous images," need to be revised to eliminate comparisons. We therefore manually check all captions and refine comparison-related captions."

**During the manual verification process, the old captions are overwritten, making it impossible to retrieve the precise before-and-after states for comparison. However, to illustrate the general concept, we can provide a hypothetical example that demonstrates the essence of the process:**

**Original description: "The fourth image features a noticeable spread of developed areas, with a larger extent than in the other images, especially strong in the top left and middle regions, indicative of a significant human footprint. Grassland areas are uniformly distributed throughout, suggesting a balance between natural landscapes and developed spaces. Crops are situated in the lower quadrants, forming large agricultural plots. Similarly, the depiction of this landscape suggests a balance between urban development and agricultural uses with some remaining grassland regions."**

**Corrected description: "This image features a noticeable spread of developed areas, especially strong in the top left and middle regions, indicative of a significant human footprint. Grassland areas are uniformly distributed throughout, suggesting a balance between natural landscapes and developed spaces. Crops are situated in the lower quadrants, forming large agricultural plots. The depiction of this landscape suggests a balance between urban development and agricultural uses with some remaining grassland regions."**

**In this example, "The fourth image" changes to "This image," "with a larger extent than in the other images," is removed, and "Similarly," is removed. This shows how the manual verification process is done by removing comparative elements. We hope this example helps clarify the manual verification process.**

3. The y-axis of Figs. 9-10 are suggested to be revised to the same to make contrasts clearer.

**R: Thank you for your valuable suggestion. We agree that using the same y-axis scale for both figures could make comparisons clearer. However, there are some considerations to keep in mind.**

**If we adjust the y-axis to the maximum frequency value of 2,000,000 for both figures, the plot for ChatGPT-4V will become difficult to interpret due to its significantly lower**

[Figure]

**frequency numbers. Alternatively, normalizing the frequency to a range of 0 to 1, as shown below, results in plots that are visually identical to Figs. 9-10 in the manuscript. However, Figs. 9-10 provide more insight into the actual number of samples in the two subsets by retaining the true value range of the frequencies.**

**While we appreciate the reviewer's suggestion, we believe that preserving the original y-axis scale provides more meaningful information and would prefer to retain the current version.**

4. The authors claim that "it stands out as the first dataset offering high-quality detailed land cover descriptions on a global scale" on line 230 of page 14. Please replace this expression with a more accurate description.

**R: Thank you for pointing out this. We have revised this sentence as follows:**

"In terms of the number of image-text pairs, the ChatEarthNet dataset is not the largest dataset available, but it offers high-quality detailed land cover descriptions on a global scale."

5. Page 10 has gaps, and the authors are encouraged to reformat the article.

**R: Thank you for the insightful suggestion. We have addressed the gaps in the layout. Please kindly check out the revised version.**

---

## Author Response (AR2)

**essd-2024-140 — Revision**

**ChatEarthNet: A Global-Scale Image-Text Dataset Empowering Vision-Language Geo-Foundation Models**

**By Z. Yuan, Z. Xiong, L. Mou, X. X. Zhu**

**General remarks to all reviewers and editors:**

**We sincerely thank the editors and anonymous reviewers for their valuable comments and suggestions. Below, we provide point-by-point responses to the reviewers' comments. The reviewers' comments are in black, and our responses follow in blue. The revised parts are marked in red in the manuscript.**

**Reviewer #1:**

I thank the authors for responding to the reviewers' comments.
I particularly appreciate the new experiments on using the dataset to evaluate vision-language models. However, there are still a few elements missing:

**R: Thank you for your thoughtful feedback on our response letter and for appreciating the additional experiments we conducted. Below, we provide detailed responses to the additional concerns raised.**

1. The new experiments (Table 2), seem to show captioning scores for several models, including GeoChat. However, there are no details as to how the setting of this evaluation. Were the same prompts used as for creating the dataset, but with the satellite image instead of the LC map?
I also miss some discussion as to why GeoChat, which has been trained with RS images, doesn't perform better than the other models. After all, one interpretation is that the proposed dataset may not be so useful for evaluating RS VLMs after all, or maybe the specific evaluation setting is not appropriate. A discussion on this, along with examples of generated captions, would be important.

**R: We appreciate the reviewer's comments and respond as follows.**

1) **To clarify, the prompt used during dataset creation and the instruction prompt for model evaluation are entirely different. When creating the dataset, we utilize land cover (LC) maps alongside carefully designed prompts to generate rich descriptions of the corresponding satellite images.**

   **However, for model training and evaluation, only satellite images are used as visual input, without any LC maps. During evaluation, all models are provided with the same instruction prompt to ensure fairness. Specifically, for the captioning task, the evaluation prompt is a variant of the instruction: "Provide a detailed description of the given image." These evaluation prompts are unrelated to the prompts designed for dataset creation.**

**To clarify this part further, we have added a description to Section 3.6 as follows:**

Note that the prompt used during dataset creation and the instruction prompt for model evaluation are entirely different. Dataset creation involves leveraging land cover maps and designing prompts to generate rich descriptions of satellite images. In contrast, during both training and evaluation of models, only satellite images are used as visual input. To ensure consistency and fairness, all models are evaluated using the same instruction prompt: "Provide a detailed description of the given image" or its variants.

2) **Thank you for highlighting this valuable question: why GeoChat [1] doesn't beat others in the zero-shot evaluation? In our zero-shot evaluation setting, GeoChat, fine-tuned on limited datasets, struggles to perform well on ChatEarthNet due to differences in spatial resolution and content coverage. However, other models are not fine-tuned on specific remote sensing datasets, which sometimes results in better generalizability. To further explain this, we add a comparison of the training datasets used by GeoChat with our ChatEarthNet dataset. GeoChat's performance on our ChatEarthNet dataset is influenced by a significant domain gap between its original training datasets and our proposed dataset. Specifically, GeoChat is trained on six datasets [2]-[7] designed for tasks like object detection, visual question answering, and scene classification on high-resolution remote sensing images, as outlined below:**

- **DOTA [2]: A dataset specifically designed for object detection in remote sensing images, with a focus on high-resolution spatial data and object categories such as ships, tennis courts, and small vehicles.**
- **DIOR [3]: Another object detection dataset with categories such as vehicle, stadium, and wind mill.**
- **FAIR1M [4]: Also an object detection dataset featuring high-resolution remote sensing imagery, providing object categories such as ship, road, and court.**
- **LRBEN [5]: A visual question answering (VQA) dataset in remote sensing, primarily addressing urban-rural classification, presence of elements (e.g., roads and buildings), and simple quantitative or comparative questions. It lacks comprehensive land use land cover (LULC) analysis.**
- **FloodNet [6]: A VQA dataset focusing on flood-related categories like flooded and non-flooded buildings or roads, with a significant domain gap from our dataset.**
- **NWPU-RESISC45 [7]: A classification dataset covering diverse scene types with varying spatial resolutions, such as bridge, church, and intersection.**

**Key differences between GeoChat's training datasets and ChatEarthNet include:**

- **Objective mismatch: GeoChat's training datasets mainly target object-centric tasks and specific queries, whereas ChatEarthNet emphasizes LULC-related semantics.**
- **Spatial resolution: Most of GeoChat's training data comprises high-resolution images focusing on objects, different from ChatEarthNet's broader geographical context.**
- **Content gap: GeoChat's training datasets lack comprehensive LULC-related descriptions required for detailed LULC analysis.**

**We add a more detailed explanation of experimental results in the Section 3.6 as follows:**

Specifically, the original GeoChat model exhibits unsatisfactory zero-shot performance on the ChatEarthNet dataset due to the substantial domain differences between its training datasets and our proposed dataset. GeoChat is trained primarily on high-resolution datasets designed for tasks such as object detection, visual question answering, and scene classification, which lack the global-scale land use and land cover-related semantics and descriptions. The differences in spatial resolution, coupled with the lack of comprehensive land cover content, significantly limit GeoChat's performance on ChatEarthNet. These gaps also motivate the need for ChatEarthNet to complement existing datasets.

3) **We would like to explain why we believe ChatEarthNet is well-suited as a benchmark for evaluating RS VLMs. RS VLMs are designed to integrate knowledge across different remote sensing data modalities and domain-specific expertise. ChatEarthNet, with its focus on global-scale LULC analysis, provides a comprehensive and challenging benchmark to test the versatility and generalization ability of these models.**

   **When fine-tuned on ChatEarthNet, models show significant performance improvements, demonstrating the dataset's ability to complement existing datasets. This makes ChatEarthNet an essential tool for assessing the breadth and depth of RS VLMs' knowledge in LULC applications on a global scale.**

4) **To provide a clearer understanding of the evaluation results, we include examples to compare the original predictions of GeoChat with those obtained after fine-tuning with ChatEarthNet. These examples demonstrate how fine-tuning the model on ChatEarthNet improves the model's ability to generate detailed and accurate descriptions. For consistency, we use the same instruction prompt: "Provide a detailed description of the given image." This comparison highlights the value of ChatEarthNet in enhancing RS VLMs' capabilities for global-scale LULC analysis.**

[Figure]

■ water ■ developed area ■ tree ▨ shrub ■ grass □ crop ▨ bare land □ snow ▨ wetland ■ mangroves ■ moss ■ unknown

**Fig. 1. Sentinel-2 satellite image and its associated land cover map. Note that during the training and evaluation process, only the sentinel-2 satellite image is used as visual input. The land cover map is only used for creating the dataset.**

**GeoChat (Original):** *In the image, there are two groups of buildings. The first group is located at the left side of the image, while the second group is located at the right side of the image. The buildings in the first group are close to each other, while the buildings in the second group are also close to each other. There are also some trees present in the image, with one tree located at the bottom right corner of the image.*

**GeoChat (w/ ChatEarthNet):** *In this image, the developed areas are scattered throughout, with a notable concentration in the central part of the image, forming irregular shapes that suggest a mix of urban and suburban layouts. The crop areas dominate the landscape, with expansive patches occupying most parts of the image, especially in the top left and bottom right sections. Small patches of grass are interspersed among the other land cover types, while the water bodies are sparse, appearing as thin, snaking lines that suggest the presence of rivers or streams. The main theme of this image is a predominantly agricultural landscape with significant urban development.*

**Label:** *In this image, the grass areas are predominant, covering a significant portion of the landscape. The developed areas have a notable presence, with an intricate network of linear and polygonal shapes indicative of urbanization. Crop areas are considerable as well, with a patchwork of rectangular and irregular shapes scattered throughout. Water bodies are visible, with sinuous and branching forms that suggest rivers or streams. The trees occupy the least area, with small, scattered clusters. Overall, the image presents a landscape dominated by grass and agricultural activities, with developed areas and water bodies interspersed, giving a sense of a mixed-use region where urban and rural elements coexist.*

[Figure]

water ▪ developed area ▪ tree ▪ shrub ▪ grass ▪ crop ▪ bare land ▫ snow ▪ wetland ▪ mangroves ▪ moss ▪ unknown

**Fig. 2. Sentinel-2 satellite image and its associated land cover map. Note that during the training and evaluation process, only the sentinel-2 satellite image is used as visual input. The land cover map is only used for creating the dataset.**

**GeoChat (Original):** *The image depicts a cityscape with a mix of buildings and trees. In the center of the image, there are two buildings close to each other. At the bottom left, there are two more buildings, and at the bottom right, there is a single building. The trees are scattered throughout the image, with some located at the bottom left, bottom right, and top right. The buildings and trees create a visually interesting and diverse scene.*

**GeoChat (w/ ChatEarthNet):** *In this image, the developed areas are predominant, with a dense network occupying a significant portion of the landscape, interspersed with patches of trees that cover varying sizes of the terrain. Small, scattered patches of grass are visible throughout, while the water bodies are minimal and irregularly shaped, mostly located near the developed areas. The main theme of this image is a landscape heavily influenced by development, with trees and grass providing some natural contrast.*

**Label:** *In this image, there is a notable dominance of tree cover in the top left, top right, and particularly in the middle part, suggesting a strong presence of forests or wooded areas. Developed regions are found throughout the image but are more concentrated in the bottom left and bottom right parts, forming patchy and fragmented patterns. A small amount of water is present, mostly noticeable in the bottom right as possibly a river or stream. Sparse patches of grass are almost imperceptible, and the bare ground is present in very small quantities. The image's landscape is primarily characterized by vast treed areas with urban development interspersed, and minimal water bodies.*

2. At the same time, the new results only aim at answering my comment 4 in the first round of review (Is the dataset useful for evaluating RS VLMs?). However, it does not deal with comment 3 (is the dataset useful for learing good RS VLM representations?). Although the authors aim at answering this question in the last two lines of Table 2, it is impossible to judge if the model is overfitting to the specific task when evaluating on the same tasks (even if on different splits). I suggest performing zero- or few-shot experiments using established RS benchmarks (like EuroSAT, BigEarthNet, etc.) in order to compare the performance of each model before and after fine-tuning with the proposed dataset.

**R: We thank the reviewer for the valuable suggestion and provide our responses below.**

1) **Following the reviewer's suggestion, we conduct additional experiments on scene classification datasets in a zero-shot setting. Specifically, we evaluate models on the widely used UCMerced [8] and AID [9] scene classification datasets. The reason we chose these two datasets is to maintain consistency with GeoChat's [1] experimental setup. Since GeoChat also uses these datasets for scene classification testing, we can directly compare our results with GeoChat's original results on the same datasets. The performance results are as follows:**

| Models | UCMerced | AID |
|---|---|---|
| Qwen-VL | 62.90 | 52.60 |
| MiniGPTv2 | 4.76 | 12.90 |
| LLaVA-1.5 | 68.00 | 51.00 |
| GeoChat | 84.43 | 72.03 |
| **GeoChat (w/ ChatEarthNet)** | **89.29** | **77.57** |

The results demonstrate that after training on the proposed ChatEarthNet dataset, the model retains its ability to perform zero-shot scene classification tasks with strong instruction-following capabilities. This highlights two key points:

- Model capacity: RS VLMs can effectively learn from diverse datasets and demonstrate their robustness and adaptability.
- No overfitting: Fine-tuning with ChatEarthNet does not lead to overfitting but instead enhances the model's representation learning, making it more capable in both detailed captioning and zero-shot scene classification tasks.

2) Regarding the concern about overfitting, we acknowledge that fine-tuning the model on a specific dataset can sometimes lead to catastrophic forgetting, a common challenge in machine learning. However, this issue is more close to model design and optimization strategies rather than the dataset itself. Our work focuses on introducing ChatEarthNet as a global-scale image-text dataset, not on developing novel VLM training methodologies. Nevertheless, our experiments show that incorporating ChatEarthNet into GeoChat's training improves performance across different tasks.

In summary, we would like to emphasize again that ChatEarthNet contributes to the remote sensing community by:

- Providing a framework for creating the large-scale and high-quality image-text dataset on remote sensing data.
- Enhancing VLMs by improving their performance when fine-tuned, without causing overfitting.

We would also like to mention that the primary aim of our manuscript aligns with the objectives of Journal *Earth System Science Data (ESSD)*: to demonstrate the process of constructing a global-scale image-text dataset using tools like ChatGPT. And the manuscript we submitted is categorized as a "Data Description" paper. While optimal training methods for RS VLMs are indeed an important topic, they fall outside the scope of this work.

We appreciate the reviewer for the deep-thought comments, and we hope these additional experiments and discussions provide a more comprehensive response to your concerns. Thank you again for your valuable suggestions.

[1] K. Kuckreja, M. S. Danish, M. Naseer, A. Das, S. Khan, and F. S. Khan, "GeoChat: Grounded large vision-language model for remote sensing," *Proc. IEEE/CVF Conf. Comput. Vision Pattern Recognit. (CVPR)*, 2024.
[2] G.-S. Xia, X. Bai, J. Ding, Z. Zhu, S. BeLongie, J. Luo, M. Datcu, M. Pelillo, and L. P. Zhang, "DOTA: A large-scale dataset for object detection in aerial images," *Proc. IEEE Conf. Comput. Vision Pattern Recognit. (CVPR)*, 2018.
[3] G. Cheng, J. Wang, K. Li, X. Xie, C. Lang, Y. Yao, and J. Han, "Anchor-free oriented proposal generator for object detection," *IEEE Transactions on Geoscience and Remote Sensing*, 2022.

[4] X. Sun, P. Wang, Z. Yan, F. Xu, R. Wang, W. Diao, J. Chen, J. Li, Y. Feng, T. Xu, M. Weinmann, S. Hinz, C. Wang, and K. Fu, "Fair1M: A benchmark dataset for fine-grained object recognition in high-resolution remote sensing imagery," *ISPRS Journal of Photogrammetry and Remote Sensing*, 2022.

[5] S. Lobry, D. Marcos, J. Murray, and D. Tuia, "RSVQA: Visual question answering for remote sensing data," *IEEE Transactions on Geoscience and Remote Sensing*, 2020.

[6] M. Rahnemoonfar, T. Chowdhury, A. Sarkar, D. Varshney, M. Yari, and R. R. Murphy, "FloodNet: A high resolution aerial imagery dataset for post-flood scene understanding," *IEEE Access*, 2021.

[7] G. Cheng, J. Han, and X. Lu, "Remote sensing image scene classification: Benchmark and state of the art," *Proceedings of the IEEE*, 2017.

[8] Y. Yang and S. Newsam, "Bag-of-visual-words and spatial extensions for land-use classification," *Proceedings of the 18th SIGSPATIAL international conference on advances in geographic information systems*, 2010.

[9] G.-S. Xia, J. Hu, F. Hu, B. Shi, X. Bai, Y. Zhong, L. Zhang, and X. Lu, "AID: A benchmark data set for performance evaluation of aerial scene classification," *IEEE Transactions on Geoscience and Remote Sensing*, 2017.

The authors answered the reviewer's questions, supplemented the experiments and made improvements to the manuscript. The reviewer recommends acceptance, but there are still two minor issues:

**R: Thank you for recommending our manuscript for acceptance. We are grateful for your acknowledgment of the efforts we made to address your comments. We have carefully considered the two minor issues you raised and provide detailed responses below.**

1. In Table A1, the authors are encouraged to unify the digital format in the "#Image-text pairs" field.

**R: We appreciate the reviewer's suggestions on this point. We have updated Table A1 to ensure a unified digital format. The revised Table A1 is presented below.**

**Table A1.** A summary of the remote sensing image-text datasets.

| Dataset | #Image-text pairs | Caption Granularity | Caption Generation | Image Data | Geographical Coverage |
|---|---|---|---|---|---|
| UCM-Captions (Qu et al., 2016) | 10,500 | Coarse-grained | Manually Annotated | RGB, UCMerced (Yang and Newsam, 2010) | Regional |
| Sydney-Captions (Qu et al., 2016) | 3,065 | Coarse-grained | Manually Annotated | RGB, Sydney (Zhang et al., 2014) | Regional |
| RSICD (Lu et al., 2017) | 54,605 | Coarse-grained | Manually Annotated | RGB, Google Earth, Baidu Map | Regional |
| NWPU-Captions (Cheng et al., 2022) | 157,500 | Coarse-grained | Manually Annotated | RGB, NWPU-RESISC45 (Cheng et al., 2017) | Regional |
| RSICap (Hu et al., 2023) | 2,585 | Fine-grained | Manually Annotated | RGB, DOTA (Xia et al., 2018) | Regional |
| RS5M (Zhang et al., 2023) |  5,000,000 | Coarse-grained | Model-generated & multiple datasets | RGB, multiple datasets | Global |
| SkyScript (Wang et al., 2024) |  2,600,000 | Coarse-grained | OpenStreetMap | RGB & multispectral, multiple sensors | Global |
| FIT-RS (Luo et al., 2024) | 1,800,851 | Fine-grained | STAR & ChatGPT | RGB, STAR (Li et al., 2024) | Global |
| RemoteCLIP (Liu et al., 2024) | 828,725 | Coarse-grained | Rule-based | RGB, multiple datasets | Global |
| ChatEarthNet | 173,488 | Fine-grained | WorldCover & ChatGPT | RGB&multispectral, Sentinel-2 | Global |

2. In Table 2, what are the specific differences between the model's original training datasets and the proposed test dataset that lead to a large gap in the metrics?

**R: Thank you for raising this insightful question. The performance gap between GeoChat's [1] original training datasets and our proposed ChatEarthNet test dataset arises due to significant domain differences. Below, we provide a detailed comparison of these domain gaps and explain why they lead to the observed performance discrepancies. Additionally, we include examples of predicted results before and after training on ChatEarthNet to better illustrate these differences. Please kindly refer to Comment#1 of Reviewer#1 for more details.**

**There is indeed a great gap between the datasets for training GeoChat and ours. GeoChat's training datasets include six datasets, summarized below:**

- **DOTA [2]: A dataset specifically designed for object detection in remote sensing images, with a focus on high-resolution spatial data and object categories such as ships, tennis courts, and small vehicles.**

- **DIOR [3]:** Another object detection dataset with categories such as vehicle, stadium, and wind mill.
- **FAIR1M [4]:** Also an object detection dataset featuring high-resolution remote sensing imagery, providing object categories such as ship, road, and court.
- **LRBEN [5]:** A visual question answering (VQA) dataset in remote sensing, primarily addressing urban-rural classification, presence of elements (e.g., roads and buildings), and simple quantitative or comparative questions. It lacks comprehensive land use land cover (LULC) analysis.
- **FloodNet [6]:** A VQA dataset focusing on flood-related categories like flooded and non-flooded buildings or roads, with a significant domain gap from our dataset.
- **NWPU-RESISC45 [7]:** A classification dataset covering diverse scene types with varying spatial resolutions, such as bridge, church, and intersection.

We summarize the following three major differences that explain the large performance gap:

- **Objective differences:** The GeoChat's training datasets are designed for tasks such as object detection, scene classification, or simple VQA, whereas ChatEarthNet emphasizes detailed LULC-related analysis and descriptions.
- **Spatial resolution:** The above-mentioned datasets predominantly feature high-resolution imagery focusing on individual objects, while ChatEarthNet provides imagery with global-scale, medium-resolution data suited for holistic LULC tasks.
- **Content and scope:** None of GeoChat's training datasets contain the comprehensive, detailed LULC-related image-text pairs that ChatEarthNet offers.

The gaps identified above demonstrate the need for ChatEarthNet, which is specifically designed to address these limitations, providing global-scale LULC-related image-text data that complements existing datasets. This enables more effective training and evaluation for applications requiring detailed LULC analysis.

We appreciate the reviewer for the deep-thought comments for improving our manuscript. We hope these additional discussions provide a comprehensive response to address your concerns. Thank you again for your valuable suggestions.

[1] K. Kuckreja, M. S. Danish, M. Naseer, A. Das, S. Khan, and F. S. Khan, "GeoChat: Grounded large vision-language model for remote sensing," *Proc. IEEE/CVF Conf. Comput. Vision Pattern Recognit. (CVPR)*, 2024.

[2] G.-S. Xia, X. Bai, J. Ding, Z. Zhu, S. BeLongie, J. Luo, M. Datcu, M. Pelillo, and L. P. Zhang, "DOTA: A large-scale dataset for object detection in aerial images," *Proc. IEEE Conf. Comput. Vision Pattern Recognit. (CVPR)*, 2018.

[3] G. Cheng, J. Wang, K. Li, X. Xie, C. Lang, Y. Yao, and J. Han, "Anchor-free oriented proposal generator for object detection," *IEEE Transactions on Geoscience and Remote Sensing*, 2022.

[4] X. Sun, P. Wang, Z. Yan, F. Xu, R. Wang, W. Diao, J. Chen, J. Li, Y. Feng, T. Xu, M. Weinmann, S. Hinz, C. Wang, and K. Fu, "Fair1M: A benchmark dataset for fine-grained object recognition in high-resolution remote sensing imagery," *ISPRS Journal of Photogrammetry and Remote Sensing*, 2022.

[5] S. Lobry, D. Marcos, J. Murray, and D. Tuia, "RSVQA: Visual question answering for remote sensing data," *IEEE Transactions on Geoscience and Remote Sensing*, 2020.

[6] M. Rahnemoonfar, T. Chowdhury, A. Sarkar, D. Varshney, M. Yari, and R. R. Murphy, "FloodNet: A high resolution aerial imagery dataset for post-flood scene understanding," *IEEE Access*, 2021.

[7] G. Cheng, J. Han, and X. Lu, "Remote sensing image scene classification: Benchmark and state of the art," *Proceedings of the IEEE*, 2017.

---

## Author Response (AR3)

**essd-2024-140 — Revision**

**ChatEarthNet: A Global-Scale Image-Text Dataset Empowering Vision-Language Geo-Foundation Models**

**By Z. Yuan, Z. Xiong, L. Mou, X. X. Zhu**

**General remarks to all reviewers and editors:**

**We sincerely thank the editors and anonymous reviewers for their valuable comments and suggestions. Below, we provide point-by-point responses to the reviewers' comments. The reviewers' comments are in black, and our responses follow in blue. The revised parts are marked in red in the manuscript.**

**Reviewer #2:**

1. In the introduction section, I recommend that the authors include a discussion of recent work, VRSBench, which provides human-verified captions rich in object details.

**R: We appreciate the reviewer's comment. We have included the discussion of the excellent recent work VRSBench in the revised introduction section as follows:**

"The recent work VRSBench [1] offers a versatile benchmark featuring human-verified captions with detailed object information for remote sensing images."

**[1] X. Li, J. Ding, and M. Elhoseiny. "VRSBench: A Versatile Vision-Language Benchmark Dataset for Remote Sensing Image Understanding." arXiv preprint arXiv:2406.12384, 2024.**

2. In Table 2, for evaluating long captions, traditional translation-based metrics can present challenges and may result in less reliable assessments. I suggest that the authors consider incorporating GPT-based metrics, such as CLAIR, to enable a more semantic-aware evaluation.

**R: Thank you for your insightful comment regarding GPT-based metrics like CLAIR for more semantic-aware assessments. We agree that CLAIR is a promising metric for such evaluations. However, its implementation requires access to the OpenAI API, which can be costly and challenging to scale for large benchmark evaluations. While using other open-source LLMs can be an alternative, the inherent limitations of these models may affect the reliability and consistency of the metric. Nevertheless, we acknowledge the potential of CLAIR and plan to explore its application in our future work.**

**While traditional metrics may have limitations in evaluating long captions, they still provide useful and standardized means to compare the performance of different models. Given the current state of research and available resources, these metrics remain a reasonable choice for benchmarking different models.**

**As the primary focus of our manuscript is on the creation of the ChatEarthNet dataset, selecting the optimal benchmarking metric falls outside the main scope of our work. However, we appreciate the reviewer's suggestion and will consider incorporating metrics like CLAIR in future studies to enhance the evaluation framework.**

**Once again, we thank the reviewer for this constructive feedback, which will help guide our future research directions.**

3. Table A1 currently lists only remote sensing image caption datasets; while image-text datasets cover more, such as VQA datasets and visual grounding datasets. The table caption should be changed.

**R: Thanks for the valuable suggestion. We have revised the caption of Table A1 and the corresponding description. The revised caption is as follows.**

**Table A1.** A summary of the remote sensing  image captioning datasets.

| Dataset | #Image-text pairs | Caption Granularity | Caption Generation | Image Data | Geographical Coverage |
|---|---|---|---|---|---|
| UCM-Captions (Qu et al., 2016) | 10,500 | Coarse-grained | Manually Annotated | RGB, UCMerced (Yang and Newsam, 2010) | Regional |
| Sydney-Captions (Qu et al., 2016) | 3,065 | Coarse-grained | Manually Annotated | RGB, Sydney (Zhang et al., 2014) | Regional |
| RSICD (Lu et al., 2017) | 54,605 | Coarse-grained | Manually Annotated | RGB, Google Earth, Baidu Map | Regional |
| NWPU-Captions (Cheng et al., 2022) | 157,500 | Coarse-grained | Manually Annotated | RGB, NWPU-RESISC45 (Cheng et al., 2017) | Regional |
| RSICap (Hu et al., 2023) | 2,585 | Fine-grained | Manually Annotated | RGB, DOTA (Xia et al., 2018) | Regional |
| RS5M (Zhang et al., 2023) | 5,000,000 | Coarse-grained | Model-generated & multiple datasets | RGB, multiple datasets | Global |
| SkyScript (Wang et al., 2024) | 2,600,000 | Coarse-grained | OpenStreetMap | RGB & multispectral, multiple sensors | Global |
| FIT-RS (Luo et al., 2024) | 1,800,851 | Fine-grained | STAR & ChatGPT | RGB, STAR (Li et al., 2024) | Global |
| RemoteCLIP (Liu et al., 2024) | 828,725 | Coarse-grained | Rule-based | RGB, multiple datasets | Global |
| ChatEarthNet | 173,488 | Fine-grained | WorldCover & ChatGPT | RGB&multispectral, Sentinel-2 | Global |

**Reviewer #3:**

**R: Thank you for recommending our manuscript for acceptance. We are grateful for your acknowledgment of our work.**